# Inhibitory medial zona incerta pathway drives exploratory behavior by inhibiting glutamatergic cuneiform neurons

Sandeep Sharma [1,2], Cecilia A. Badenhorst[1,2], Donovan M. Ashby[1,3], Stephanie A. Di Vito [1], Michelle A. Tran [1,2], Zahra Ghavasieh[1], Gurleen K. Grewal[1], Cole R. Belway [1], Alexander McGirr [1,3] & Patrick J. Whelan [1,2] ✉

The cuneiform nucleus (CnF) regulates locomotor activity, which is canonically viewed as being primarily involved in initiating locomotion and regulating speed. Recent research shows greater context dependency in the locomotor functions of this nucleus. Glutamatergic neurons, which contain vesicular glutamate transporter 2 (vGLUT2), regulate context-dependent locomotor speed in the CnF and play a role in defensive behavior. Here, we identify projections from the medial zona incerta (mZI) to CnF vGLUT2 neurons that promote exploratory behavior. Using fiber photometry recordings in male mice, we find that mZI gamma-aminobutyric acid (GABA) neurons increase activity during periods of exploration. Activation of mZI GABAergic neurons is associated with reduced spiking of CnF neurons. Additionally, activating both retrogradely labeled mZI-CnF GABAergic projection neurons and their terminals in the CnF increase exploratory behavior. Inhibiting CnF vGLUT2 neuronal activity also increases exploratory behavior. These findings provide evidence for the context-dependent dynamic regulation of CnF vGLUT2 neurons, with the mZI-CnF circuit shaping exploratory behavior.

The survival of an organism depends on its ability to sense, process, and respond adaptively to stimuli (internal or external) in order to engage in behaviors such as feeding, finding a mate, defensive behavior, or exploring a habitat. Integrating higher brain centers with highly conserved descending brainstem locomotor command centers is key in coordinating context-specific locomotor behaviors[1–4]. The mesencephalic locomotor region (MLR) is one such conserved descending locomotor center, which is present in diverse vertebrate species known to produce locomotion[5–9]. MLR neurons are spontaneously active and under tonic inhibitory control from several subcortical areas, including the substantia nigra pars reticulata[10] and basal ganglia[10,11]. Disinhibition of MLR neurons is thought to drive spontaneous locomotion[12]. Therefore, understanding how inhibitory descending control is modulated is critical to understanding MLR output.

The cuneiform nucleus (CnF) and pedunculopontine tegmental nucleus (PPN), both part of the MLR, have differing roles in locomotor control. Early studies demonstrated that stimulation of the MLR produces a range of locomotor responses that could support escape behavior, including explosive high-speed running[13]. Excitatory glutamatergic CnF neurons containing vesicular glutamate transporter 2 (vGLUT2) set both the speed and gait that is required for locomotion[14–16]. In turn, chemogenetic inhibition of CnF vGLUT2 neurons reduces the ability to perform high-speed gaits required for escape, but preserves slow gaits[14]. Gamma-aminobutyric acid (GABA) neurons are also found within the CnF and stimulation of CnF GABAergic neurons is associated with the stopping of locomotion[14].

The PPN is a more heterogeneous structure, consisting of cholinergic, glutamatergic, and GABAergic neurons. The inputs and

[1]Hotchkiss Brain Institute, University of Calgary, Calgary, AB T2N 4N1, Canada. [2]Faculty of Veterinary Medicine, University of Calgary, Calgary, AB T2N 4N1, Canada. [3]Department of Psychiatry, Cumming School of Medicine, University of Calgary, Calgary, AB T2N 4N1, Canada. ✉e-mail: whelan@ucalgary.ca

outputs of the PPN are more diverse than those of the CnF[14,16], contributing to a wide range of functions. These include location-dependent activation of motor behavior[14–18], and effects on arousal, sleep, and autonomic function[19–21]. However, the CnF has connectivity with the sympathetic nervous system[22–25], which leads to increased blood pressure and respiratory function[26]. As a result, the CnF likely produces preparatory increases in sympathetic tone to support initiation and maintenance of locomotion. Less is known about the dynamic regulation of the CnF during other goal-directed locomotor behaviors like foraging and exploratory behavior. These complex behaviors are integrated into the emotional states of the brain[1], suggesting a possible role for the CnF.

The CnF receives major inputs from the basal ganglia and the zona incerta (ZI)[14,16]. The ZI has been implicated in multiple functions including: feeding[27], sexual function[28], autonomic regulation[29], attention[30], postural control[31], and locomotion[32–34]. The mZI is a region of interest because the ZI is recognized as a hub for sensorimotor integration and is associated with the genesis of complex locomotor behaviors[34–38]. Recent work has shown that GABAergic neurons in the ZI contribute to appetitive drive leading to hunting behaviors[35]. Others have reported that GABAergic ZI neurons produce defensive behaviors[37,39]. Moreover, ZI projections to the periaqueductal gray (PAG) are involved in the extinction of defensive behaviors[37], while the activation of ZI GABAergic inputs to the PPN can suppress active avoidance behaviors[34]. Overall this suggests a high degree of specialization within the ZI, and that circuit-specific control of the CnF from the ZI may have an important role in controlling behavior. Anatomically, the ZI is divided into four quadrants[38,40]. Here, we examine the effects of inhibiting CnF vGLUT2 neurons, which triggers episodes of exploratory activity. We find that during exploratory behaviors, mZI GABA neurons are active. In support of this data, we also find that activating GABAergic inputs from the mZI onto the CnF are sufficient to induce exploratory behaviors. Altogether, this study shows the presence of an inhibitory pathway extending from the mZI to the CnF that promotes exploratory behavior. These findings expand our understanding of the mZI, as a region that integrates multimodal sensory input[38,41], including modulatory control of the CnF.

## Results

### Photoinhibition of CnF vGLUT2 neurons promotes exploratory behavior

We first addressed the functional role of photoinhibition of CnF vGLUT2 neurons in freely moving mice and whether this would suppress ongoing locomotor activity and speed of locomotion. A Cre-dependent adeno-associated virus (AAV) expressing halorhodopsin with enhanced yellow fluorescent protein (AAV-DIO-eNpHR3.0-eYFP) was injected into the CnF of vGLUT2-IRES-Cre mice (vGLUT2[eNpHR3.0]) and an optic fiber was implanted above the CnF (Fig. 1a, b). Littermates injected with AAV-DIO-eYFP (vGLUT2[eYFP]) were used as controls. Next, movement and spontaneous locomotor activity were assessed in the open field test in animals expressing CnF vGLUT2[eNpHR3.0] or vGLUT2[eYFP]. Baseline spontaneous locomotor activity was recorded for 3 minutes before yellow laser light onset, directly followed by two successive epochs of 3 min of continuous yellow laser light and 3 min of post-light recording (Fig. 1c). Widely reported measures of spontaneous locomotor activity in the open field test, including distance traveled, speed, immobility, visits to the center zone, and rearing are interpreted to measure exploratory drive of the rodents. In the open field environment, mice tend to exhibit limited behaviors including immobility, rearing, and locomotion. These behaviors were displayed using ethograms (Fig. 1e; Supplementary Fig. 1). Photoinhibition of CnF vGLUT2 neurons significantly reduced immobility duration in vGLUT2[eNpHR3.0] mice during laser ON epoch (Fig. 1f). We observed that the speed of locomotion in both vGLUT2[eYFP] and vGLUT2[eNpHR3.0] mice was restricted

to $9.1 \pm 0.4$ cm/s and $9.8 \pm 0.6$ cm/s, respectively, and photoinhibition did not affect locomotor speed (Fig. 1g). Changes in preference for thigmotactic areas (walls and corners) or the brightly lit central zone are used to measure and report the exploratory drive of rodents in the open field test. Interestingly, we observed an increase in the center distance traveled (Fig. 1i) and the time spent in the center zone (Fig. 1j) in vGLUT2[eNpHR3.0] compared to control vGLUT2[eYFP] mice. *Post hoc* analysis showed that the vGLUT2[eNpHR3.0] mice had an increased preference for the center zone under yellow light inhibition (Fig. 1j). The preference for the center zone appeared higher than the baseline during laser OFF conditions but did not approach significance ($p = 0.07$, Fig. 1j) in vGLUT2[eNpHR3.0] mice.

These results suggest an increase in exploratory behavior, the specific form of which may be predicated on decreased anxiety, under inhibition of CnF vGLUT2 neurons. To examine this possibility, we tested mice in the Elevated Plus Maze, a widely used behavioral assay to measure anxiety-like and exploratory behaviors[42] (Fig. 2a). We observed an increase in both the visits (Fig. 2b) and the time spent in the open arms (Fig. 2c) of the elevated plus maze in vGLUT2[eNpHR3.0] mice compared to vGLUT2[eYFP] mice during photoinhibition. Next, we tested these mice in the light-dark transition test to compare their natural aversion to bright light (Fig. 2d). Our results show an increase in the time spent in the light chamber in vGLUT2[eNpHR3.0] mice compared to the control vGLUT2[eYFP] mice during photoinhibition (Fig. 2e). Moreover, time spent per visit to the light chamber increased in vGLUT2[eNpHR3.0] mice (Fig. 2f).

We next used a hole-board arena to examine head-dips, which are associated with exploratory behavior (Fig. 2g). Our results show that photoinhibition increased the frequency of head-dips in vGLUT2[eNpHR3.0] mice (Fig. 2h) whereas these effects were not observed in the control vGLUT2[eYFP] mice (Fig. 2h). We also observed that photoinhibition increased the duration of head-dips in vGLUT2[eNpHR3.0] mice (Fig. 2i) whereas these effects were not observed in the control vGLUT2[eYFP] mice.

Collectively, these results from the open field test, elevated plus maze, light-dark transition test, and hole-board test suggest an increase in exploratory behavior associated with photoinhibition of CnF vGLUT2[eNpHR3.0] neurons.

### mZI acts as a source of inhibitory projections onto CnF vGLUT2 neurons

To map the source of inhibitory GABAergic projections to the CnF, we injected a Cre-dependent retrograde AAV reporter fluorescent protein (AAVrg-DIO-ChR2-eYFP) into the CnF of vesicular GABA transporter (vGAT)-IRES-Cre mice (Fig. 3a). Consistent with other reports, cell bodies were found in the superior and inferior colliculus, PAG, amygdala, and the mZI[14,43]. Monosynaptic retrograde studies have shown the presence of projection neurons in ZI targeting MLR vGLUT2 neurons[14,16]. However, the mZI functional connectome to the MLR is not well understood. The mZI is predominantly GABAergic and innervates midbrain areas including the superior colliculus, PAG, and PPN[34,44]. We injected a Cre-dependent anterograde AAV (AAV-DJ-hSyn-Flex-mGFP-2A-Synaptophysin-mRUBY) into the mZI of vGAT-IRES-Cre mice to label projection terminals with synaptophysin-mRUBY (SYP-mRUBY[+], Fig. 3b). We confirmed the presence of SYP-mRUBY[+] puncta in previously reported target regions of mZI GABAergic neurons i.e., PAG and PPN[34,44] (Supplementary Fig. 2a, b). Our results confirmed the presence of mZI GABAergic synaptic terminals in the CnF (Fig. 3c). Anterograde tracing results generally supported our retrograde findings, showing ipsilateral mZI GABAergic synaptic puncta in the CnF. However, sparse contralateral CnF projections were found, suggesting a unilateral biased projection overall for the mZI-CnF circuit (Supplementary Fig. 2c).

To confirm that mZI GABAergic neurons form putative synaptic contacts with vGLUT2 neurons, we injected the Cre-dependent

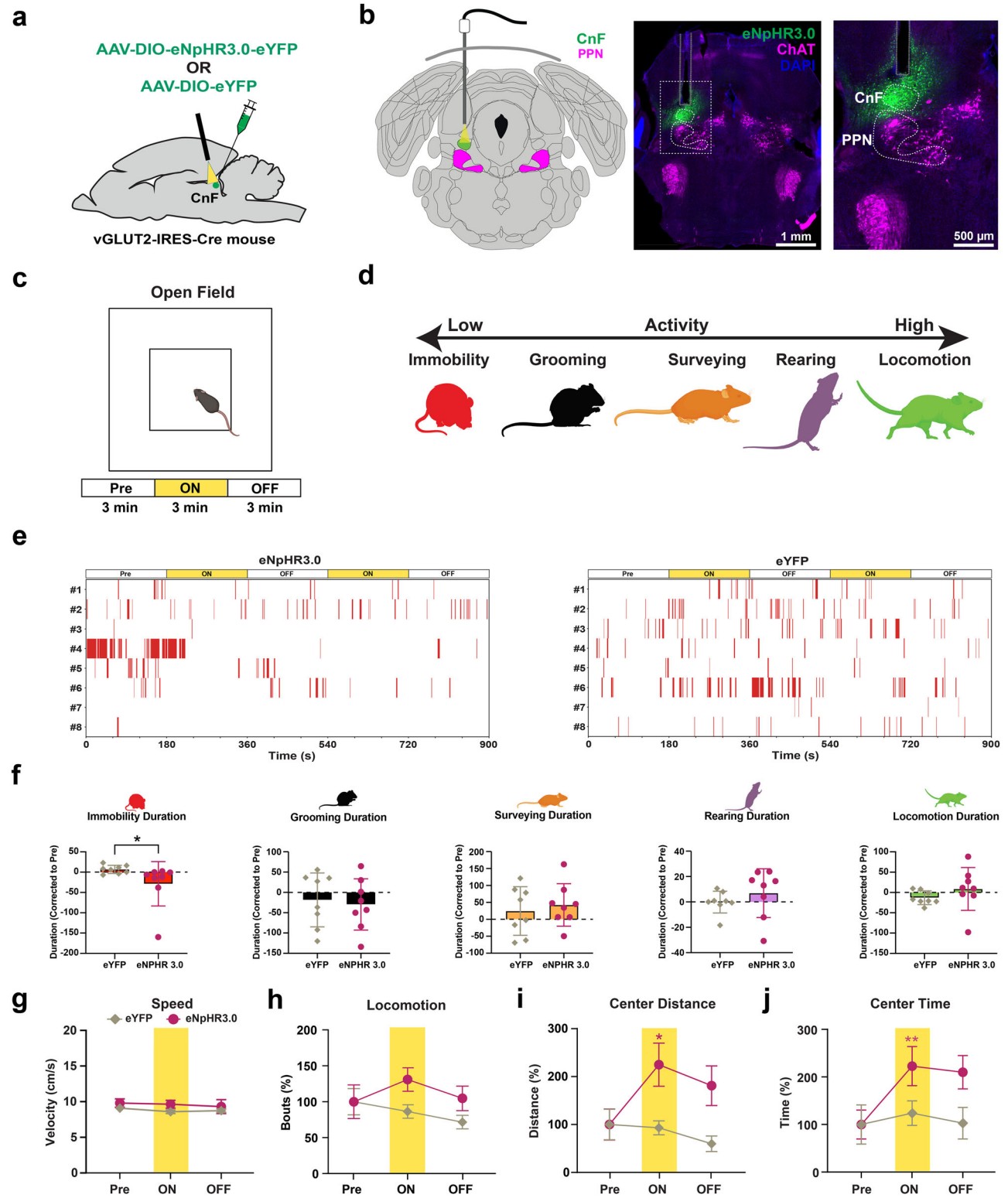

anterograde AAV (AAV-DJ-hSyn-Flex-mGFP-2A-Synaptophysin-mRUBY) into the mZI in a separate cohort of vGAT-IRES-Cre mice (Fig. 3d). We used RNAscope® to identify CnF cells with vGLUT2 mRNA. Our results revealed that the majority of CnF cells receiving putative innervation from the mZI, indicated by SYP-mRUBY[+] synaptic puncta, are vGLUT2[+] cells (Fig. 3e). These data also indicate that a smaller proportion of mZI GABAergic afferents project to CnF vGLUT2[-] cells, presumably vGAT neurons. SYP-mRUBY[+] synaptic puncta were evaluated across all vGLUT2[+] cells within the CnF. Our results show that

there is a relatively smaller proportion of CnF vGLUT2[+] cells that are associated with SYP-mRUBY[+] synaptic puncta (Fig. 3e).

The anatomical data suggested that putative GABAergic synapses from mZI were present in the CnF, but their functional effect was not established. Therefore, we tested whether photo-stimulation of mZI vGAT[ChR2] populations leads to a decrease in the CnF cells firing rates. In anesthetized mice, we found that photo-stimulation of GABAergic somata in the mZI led to an inhibition of firing frequency at short (<100 ms) latencies in a subpopulation of

**Fig. 1 | Inhibition of CnF vGLUT2 neurons increases exploratory locomotor behavior in the open field. a, b** Schematic depicts viral injection and photo-inhibition of the CnF vGLUT2 neurons; representative images display vGLUT2 neurons (green) with an optic fiber above the CnF (eNpHR3.0: $n = 8$ mice; eYFP: $n = 8$ mice). **c, d** Schematic illustrates open field locomotor activity testing proto-col, with events classified as immobility (red), grooming (black), surveying (orange), rearing (purple), and locomotion (green). **e** Ethograms show immobility events during 3 minute epochs before (Pre), during (ON), and after (OFF) yellow laser light (561 nm). **f** Panel shows total duration of behaviors during stimulation across trials after baseline correction to pre-stim conditions. Photoinhibition of CnF vGLUT2 neurons significantly reduced the immobility duration (eNpHR3.0: $n = 8$ mice; eYFP: $n = 8$ mice, $U = 8.50$, $p = 0.0115$, two-sided Mann–Whitney test, data are presented as mean ± SEM) in vGLUT2^eNpHR3.0 mice during laser ON epoch.

**g** Photoinhibition of CnF vGLUT2 neurons did not alter locomotion speed. **h** Total number of locomotion bouts are unchanged in vGLUT2^eNpHR3.0 mice compared to the vGLUT2^eYFP mice. **i, j** An increase in the locomotor center distance traveled (eNpHR3.0: $n = 8$ mice; eYFP: $n = 8$ mice, Time: $F_{(1.32,18.55)} = 4.37$, $p = 0.041$; Virus × Time: $F_{(2,28)} = 6.53$, $p = 0.0047$; post hoc vGLUT2^eNpHR3.0 Pre vs light ON, $p = 0.027$, Tukey's multiple comparisons test) and the time spent in the center of the open field (eNpHR3.0: $n = 8$ mice; eYFP: $n = 8$ mice, $Q = 9.75$, $p = 0.0048$, Friedman test; post hoc vGLUT2^eNpHR3.0 Pre vs light ON, $Z = 3.00$, $p = 0.0081$; Dunn's multiple comparisons test) was observed in vGLUT2^eNpHR3.0 mice in laser ON condition. Two trials per mouse were conducted. The data were normalized to baseline Pre con-dition within each group and presented as mean ± SEM. Atlas images adapted from the Allen Mouse Brain Atlas[68,69]. Illustrations in **a–d, f** created with BioRender.com with permission. *$p < 0.05$, **$p < 0.01$.

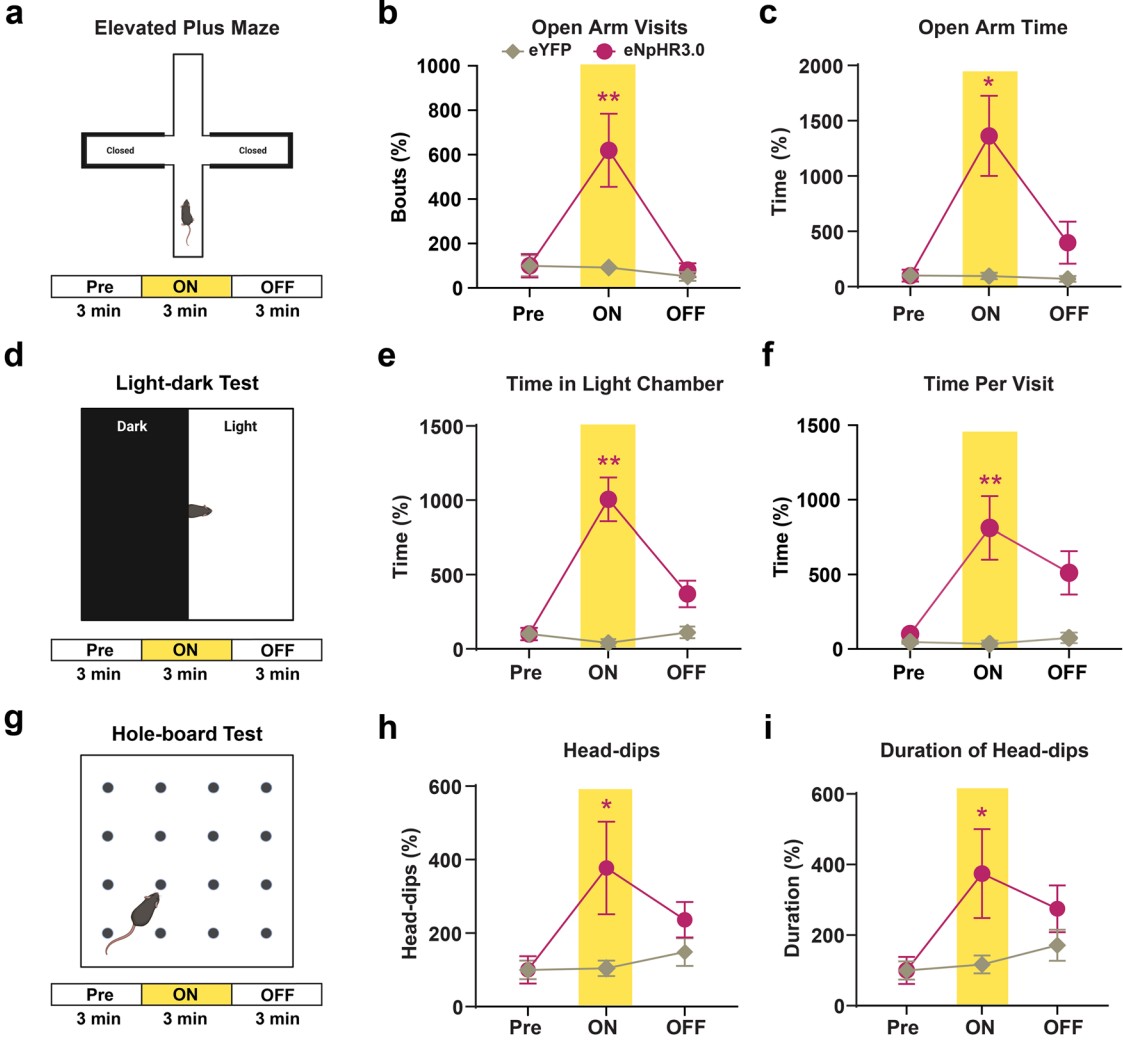

**Fig. 2 | Inhibition of CnF vGLUT2 neurons produces exploratory behavior.**
**a** Illustration shows elevated plus maze testing paradigm. **b, c** An increase in both the open arm visits (eNpHR3.0: $n = 9$ mice; eYPF: $n = 9$ mice, $Q = 14.47$, $p = 0.0003$, Friedman test; post hoc vGLUT2^eNpHR3.0 Pre vs light ON, $Z = 2.95$, $p = 0.0096$; vGLUT2^eNpHR3.0 ON vs light OFF, $Z = 3.064$, $p = 0.0065$; Dunn's multiple comparisons test) and the time spent in the open arms ($Q = 13$, $p = 0.0004$, Friedman test; post hoc vGLUT2^eNpHR3.0 Pre vs light ON, $Z = 3.3$, $p = 0.0029$; Dunn's multiple comparisons test) was observed in vGLUT2^eNpHR3.0 mice during laser ON conditions. **d** Illustration shows light-dark transition test arena testing paradigm. **e, f** An increase in both the time spent in the light chamber (eNpHR3.0: $n = 8$ mice; eYFP: $n = 6$ mice, $Q = 12.25$, $p = 0.0009$, Friedman test; post hoc vGLUT2^eNpHR3.0 Pre vs light ON, $p = 0.0014$, Dunn's multiple comparisons test) and the time per visit in the light chamber

($Q = 9.750$, $p = 0.0048$, Friedman test; post hoc vGLUT2^eNpHR3.0 Pre vs light ON, $p = 0.0081$, Dunn's multiple comparisons test) occurred in vGLUT2^eNpHR3.0 mice during laser ON epoch. **g** Illustration shows hole-board testing paradigm. **h, i** Both the number of head-dips (eNpHR3.0: $n = 10$ mice; eYFP: $n = 9$ mice, $Q = 8.05$, $p = 0.015$, Friedman test; post hoc vGLUT2^eNpHR3.0 Pre vs light ON, $Z = 2.80$, $p = 0.016$; Dunn's multiple comparison test) and their duration ($Q = 8.60$, $p = 0.012$, Friedman test; post hoc vGLUT2^eNpHR3.0 Pre vs light ON, $Z = 2.91$, $p = 0.011$, Dunn's multiple comparisons test) show a significant increase in vGLUT2^eNpHR3.0 mice during pho-toinhibition. The data are normalized to baseline Pre condition within each group and presented as mean ± SEM. Illustrations in **a, d, g** created with BioRender.com. *$p < 0.05$, **$p < 0.01$.

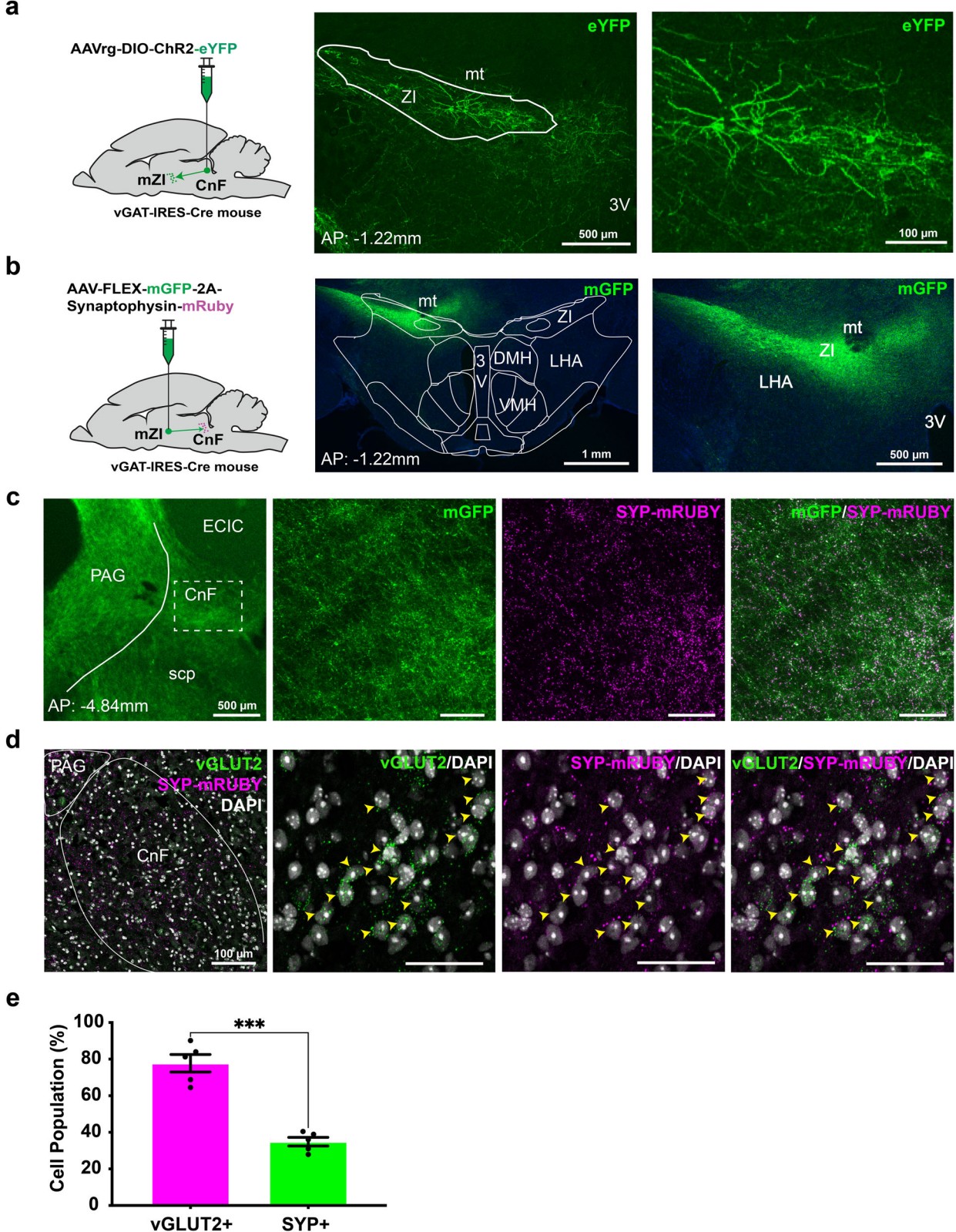

CnF cells (Fig. 4). This inhibition occured in neurons with both low and high tonic spiking (baseline firing rate range 2.2–31.8 Hz, mean $16.0 \pm 4.5$ Hz). Consistent with our anatomical data showing a subpopulation of vGLUT2 neurons with SYP-mRUBY⁺ puncta (Fig. 3e), our electrophysiological data show 26% of units (7 of 27, Fig. 4a–f, $p < 0.05$) were inhibited following mZI photostimulation, showing a statistically significant change from the baseline rate in

response to a 5 pulse optogenetic stimulation. No units showed a statistically different increase from the baseline rate in response to photostimulation (27 of 27, $p > 0.1$). Collectively, these data confirm the presence of an anatomical mZI-CnF circuit that provides inhibitory input to CnF vGLUT2 cells and indicate that a subpopulation of CnF cells are functionally inhibited following mZI GABAergic photostimulation.

**Fig. 3 | Existence of an mZI-CnF inhibitory circuit innervating CnF vGLUT2 neurons. a** Schematic showing injection of the retrograde Cre-dependent virus, AAVrg-DIO-ChR2-eYFP, in the CnF of vGAT-IRES-Cre mice (*n* = 2 mice). Representative images show Cre-dependent expression of reporter in the mZI cells (in green). **b** Schematic showing mZI injection of the anterograde Cre-dependent virus AAV-Flex-mGFP-2A- Synaptophysin-mRuby labeling synaptic puncta in vGAT-IRES-Cre mice (*n* = 2 mice). Representative images show the efficacy of viral expression in mZI (green). **c** Representative confocal images show mZI inhibitory fibers in the CnF (green) and synaptic puncta (magenta). The CnF region, outlined by the dotted rectangle, is enlarged to show the presence of mZI inhibitory fibers (green) and the inhibitory mZI synaptic puncta (magenta). **d** Representative confocal images (*n* = 5

mice) show the presence of mZI inhibitory synaptic puncta (SYP-mRuby, magenta) on the CnF vGLUT2 neurons (green) labeled with arrowheads. **e** A majority of the SYP-CnF cells are vGLUT2$^+$ (magenta bar); while a lower proportion of the CnF vGLUT2$^+$ cells are SYP$^+$ (green bar, *n* = 5 mice, $t_{(5)}$ = 8.009, *p* = 0.0002, two-sided unpaired t-test with Welch's correction). Data are presented as mean ± SEM. Scale bars are 50 μm, unless otherwise indicated. Atlas images adapted from the Allen Mouse Brain Atlas[68,69]. Illustrations in **a** and **b** created with BioRender.com. mt mammillothalamic tract, 3V third ventricle, DMH dorsomedial hypothalamus, VMH ventromedial hypothalamus, LHA lateral hypothalamic area, ECIC external cortex of the inferior colliculus, scp superior cerebellar peduncle. **\*\****p* < 0.01, **\*\*\****p* < 0.001.

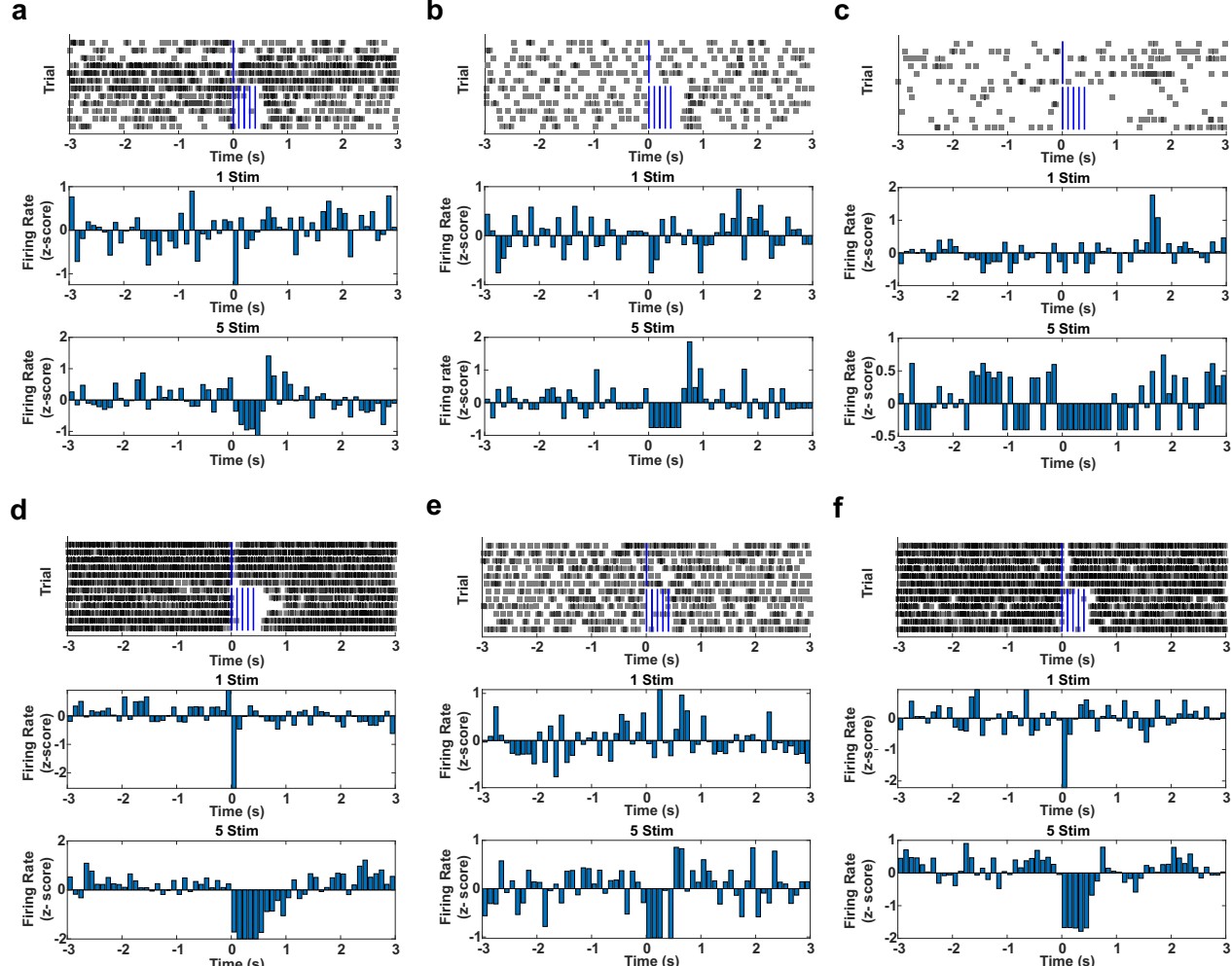

**Fig. 4 | CnF neurons are inhibited following mZI GABAergic photostimulation. a–f** Cells recorded in the CnF responding to optogenetic stimulation of the mZi vGAT neurons (*n* = 27 cells from 3 mice). Top panels show raster plots of individual spikes over 6 single-pulse trials followed by 6, 5-pulse burst trials. Middle panels

show firing rates (expressed as z-scores) in 100 ms bins averaged across the 6 single pulse trials. Bottom panels show firing rates (expressed as z-scores) in 100 ms bins averaged across the 6, 5-pulse trials.

## mZI inhibitory GABAergic neurons are active during exploratory behavior

Next, we investigated whether mZI GABAergic neuronal activity was associated with exploratory behavior. We injected vGAT-IRES-Cre mice with a Cre-dependent AAV construct containing GCaMP6s (AAV9-CAG.Flex.GCaMP6s) into the mZI (vGAT$^{GCaMP6s}$, Fig. 5a) and implanted an optic fiber over the mZI to record calcium (Ca$^{2+}$) changes in vGAT$^{GCaMP6s}$ neurons in freely behaving mice (Fig. 5b, c). Head-dipping activity has been widely utilized as an indicator of exploratory behavior in rodents[14,45,46]. Thus, we tested exploratory behavior using the hole-

board test[14]. Immobility (Fig. 5d), rearing (Fig. 5e), and head-dipping (Fig. 5f) all require mice to be stationary, but since we observed an increase in Ca$^{2+}$ transients only during the onset of rearing and head-dipping, this suggests a role for mZI GABAergic neurons in exploratory behavior.

To assess changes in Ca$^{2+}$ transients in mZI vGAT$^{GCaMP6s}$ neurons during locomotion, we categorized locomotion bouts into two distinct behavioral types. First, we identified thigmotactic locomotor bouts where mice walk around the edges of the hole-board arena (Fig. 5g, left). We observed that Ca$^{2+}$ transients in mZI vGAT$^{GCaMP6s}$ neurons were

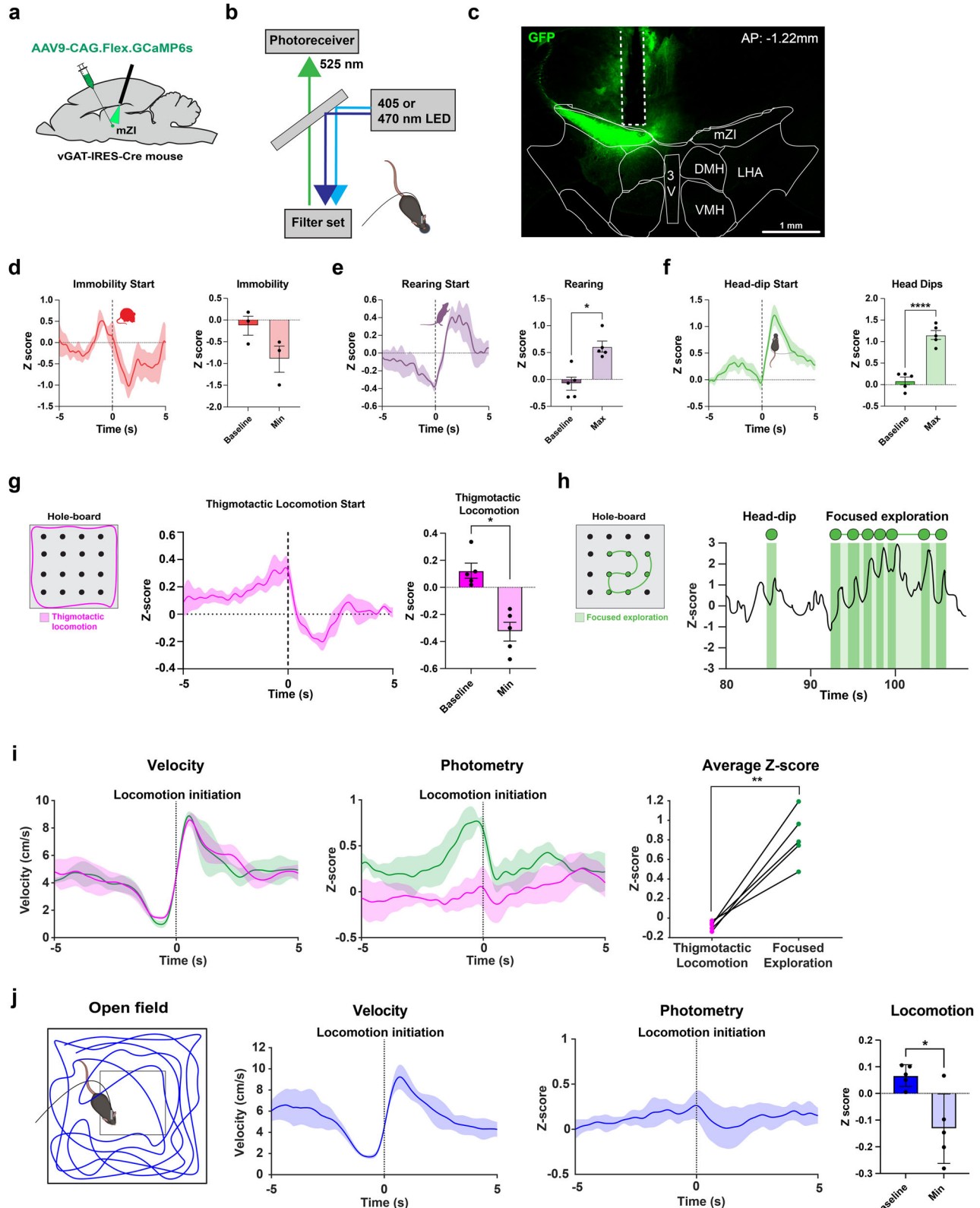

reduced at the onset of thigmotactic locomotion (Fig. 5g, right) suggesting an inverse relationship between mZI GABAergic neuronal activity and thigmotactic locomotion. Head-dips in the holes are considered to be part of focused exploration[47]. Focused exploration is defined when a mouse moves from hole-to-hole, engaging in spontaneous head-dipping behavior at each hole in quick succession (Fig. 5h, left). Figure 5h (right) displays a representative trace of a mouse

exhibiting both isolated head-dips and focused exploration. We identified locomotor bouts associated with focused exploration and compared them with thigmotactic locomotor bouts occurring during the same session (Fig. 5i). Ca²⁺ transients in mZI vGAT^GCaMP6s neurons significantly increased during focused exploration compared to thigmotactic locomotion (Fig. 5i, right). Consistent with the increased mZI GABAergic neuronal activity during focused exploration (Fig. 5i), we

**Fig. 5 | mZI inhibitory vGAT neurons are active during exploratory behavior.** **a**–**c** Schematics show viral injection and mZI vGAT neuronal activity recording using single-fiber photometry; representative image displays expression of GCaMP6s in mZI vGAT neurons (green) with the ferrule track implanted above ($n = 5$ mice). **d**–**f** Averaged traces of mZI vGAT neuronal activity during episodes of immobility (red, $n = 3$ mice), rearing (purple, $n = 5$ mice) and head-dips (green, $n = 5$ mice) in the hole-board test. Quantified difference in z-score values from baseline for **d**, immobility ($n = 5$ mice, $p = 0.223$, $t_{(4)} = 1.745$), e, rearing ($n = 5$ mice, $t_{(4)} = 4.165$, $p = 0.0141$, two-tailed paired t-test) and f, head-dips ($n = 5$ mice, $t_{(4)} = 19.19$, $p < 0.0001$, two-tailed paired t-test) showed an increase in neuronal activity. **g** mZI vGAT neuronal activity decreased after the onset of thigmotactic locomotion (pink) ($n = 5$ mice, $t_{(4)} = 4.085$, $p = 0.0150$, two-tailed paired t-test). **h** Representative trace showing mZI vGAT neuronal activity during an isolated head-dip and focused exploration (in green) involving successive head-dips in quick succession with locomotion between holes. **i** Similar velocity of thigmotactic locomotion (pink) and focused exploration locomotion (green) aligned at the initiation of locomotion. Increase in $Ca^{2+}$ transients of mZI vGAT neurons during focused exploration compared to thigmotactic locomotion ($n = 5$ mice, $t_{(4)} = -7$, $p = 0.0022$, two-tailed paired t-test). **j** Velocity of locomotion (blue) aligned at the initiation of locomotion in the open field. $Ca^{2+}$ activity significantly decreased after the onset of locomotion in the open field ($n = 5$ mice, $t_{(4)} = 3.638$, $p = 0.0222$, two-tailed paired t-test). Data are presented as mean ± SEM. Atlas image adapted from the Allen Mouse Brain Atlas[68,69]. Illustrations in **a**, **b**, **d**–**h**, **j** created with BioRender.com. 3V third ventricle, DMH dorsomedial hypothalamus, VMH ventromedial hypothalamus, LHA lateral hypothalamic area. **$**p < 0.01$.

also observed an increase in $Ca^{2+}$ transients at the onset of head-dips and this activity peaked at the end of a head-dip (Fig. 5f). Unlike focused exploration, at similar velocities, thigmotactic locomotion was associated with relatively modest changes in $Ca^{2+}$ activity (Fig. 5i). Next, we examined changes in $Ca^{2+}$ transients during activity in the elevated plus maze. We observed significantly increased $Ca^{2+}$ transients associated with excursions from the closed arm to the center and open arms in the elevated plus maze (Supplementary Fig. 3), consistent with increased exploratory behavior and anxiolysis.

The open field behavioral repertoire lacks behaviors associated with focused exploration seen in the hole-board. Interestingly, $Ca^{2+}$ transients showed a modest decrease following locomotion initiation during the open field test (Fig. 5j). These results highlight a role of ZI GABAergic neuronal activity in context-specific exploratory behaviors, particularly when mice decrease their speed and are engaged in exploratory behaviors.

### mZI promotes exploratory behavior via inhibitory projections to the CnF

While activating mZI GABAergic inputs to the PPN can suppress active avoidance behaviors[34], circuit-specific modulation of inhibitory neurons onto the CnF is not well understood. We next tested a cohort of vGAT-IRES-Cre mice that were injected with either AAV-DIO-ChR2-eYFP or control AAV-DIO-eYFP into the mZI, and implanted with an optic fiber above the CnF to target mZI-CnF GABAergic terminals (Fig. 6a, b). Mice were tested in the open field to measure spontaneous locomotor behavior (Fig. 6c). We cataloged mouse behavioral activity (Fig. 6d) as rearing, locomotion, surveying, grooming and immobility in ethograms (Fig. 6e, f, Supplementary Fig. 4). Photostimulation of mZI GABAergic terminals in vGAT[ChR2] mice decreased the proportions of immobility and increased rearing compared to vGAT[eYFP] mice during laser ON condition (Fig. 6g). We did not observe any significant change in locomotor speed or the locomotor bouts in vGAT[ChR2] mice during photostimulation (Fig. 6h, i). However, we observed a significant increase in both the center distance traveled (Fig. 6j) and the time spent in the center zone of the open field in vGAT[ChR2] mice during photostimulation (Fig. 6k). These results suggest that photostimulation evoked increased exploratory behavior in the open field.

Next, we tested these mice for exploratory behavior and anxiety-like behavior in the elevated plus maze (Fig. 7a). We found that during photostimulation, there was a significant increase in the visits to the open arms (Fig. 7b) and the time spent in the open arms of the elevated plus maze in vGAT[ChR2] mice (Fig. 7c). We next tested whether photostimulation during the light-dark transition test produced changes in preference for the light chamber, suggesting an increase in exploratory drive (Fig. 7d). Photostimulation of mZI GABAergic terminals in the CnF significantly increased preference for the light chamber in the light-dark transition test in vGAT[ChR2] mice (Fig. 7e) with a significant increase in the duration per visit to the light chamber (Fig. 7f). To further evaluate exploratory drive, we photostimulated mice in the hole-board arena (Fig. 7g). During photostimulation, we found that

mice showed an increase in both the number of head-dips (Fig. 7h) and the duration of head-dips (Fig. 7i).

To further support that mZI-CnF circuit contributed to exploratory locomotor behavior, we injected a retrograde AAVrg-DIO-ChR2-eYFP into the CnF of vGAT-IRES-Cre mice and implanted an optic fiber above the mZI (Fig. 8a). We found that a subpopulation of neurons in the mZI were labeled, and that labeled soma were largely ipsilateral to the retrograde injection (Fig. 8a). Upon photostimulation of the ChR2 in retrogradely labeled mZI neurons, we found that the elevated plus maze data showed no significant change in behavior (Fig. 8c,d). In the light-dark transition test, mice spent more time in the light chamber after photostimulation (Fig. 8f), while the time per visit remained unchanged (Fig. 8g). Finally, the head-dips (Fig. 8i) and duration of head-dips (Fig. 8j) were significantly increased in the hole-board test. Overall, photostimulation of the mZI GABAergic neurons projecting to the CnF produced similar, but attenuated results, compared to the mZI-CnF GABAergic terminal photostimulation experiments.

## Discussion

Our work examined the effects of inhibiting CnF vGLUT2 neurons and demonstrated how this can trigger episodes of exploratory activity. GABAergic inputs from the mZI onto CnF vGLUT2 cells are sufficient to trigger this behavioral switch. Furthermore, we found that mZI GABA neurons are active during exploratory behaviors, which is consistent with recent reports showing the presence of inhibitory neuron subpopulations in the ZI that are implicated in novelty processing and anxiolysis[48]. Finally, we provide evidence of a decrease in CnF neuron firing rates following mZI photostimulation, consistent with an increase in mZI $Ca^{2+}$ transients, during decreases in locomotor velocity. Collectively, this suggests circuit-specific control from the mZI, an area that integrates multimodal sensory input and drives modulatory control of the CnF.

Several lines of evidence suggest both functional and anatomical links between mZI and CnF. Our anatomical evidence shows inhibitory mZI vGAT projections and their presynaptic terminals labeled as synaptophysin-positive boutons, in proximity with CnF vGLUT2 neurons. Furthermore, extracellular recordings from CnF neurons show inhibition of firing upon photostimulation of vGAT neurons in the mZI. This adds to our behavioral data showing the efficacy of photostimulation of GABAergic mZI projecting fibers in the CnF. Previous research using cell-type specific methods have found that CnF vGLUT2 neurons play a role in generating a wide range of gaits such as walk, gallop and bound, and can induce these gaits independently of PPN vGLUT2 neurons[14–17]. Indeed, activity in PPN vGLUT2 neurons promotes slow-speed movements for exploratory behavior[14–16]. Excitatory vGLUT2 neurons in the CnF and the PPN have reciprocal projections, with dominant projections from the CnF to the PPN[14,16]. These provide potential circuits for CnF vGLUT2 neurons to modulate PPN neurons in the range of slower walking gaits. In our work, inhibition of CnF vGLUT2 neurons in mice performing spontaneous locomotion in the

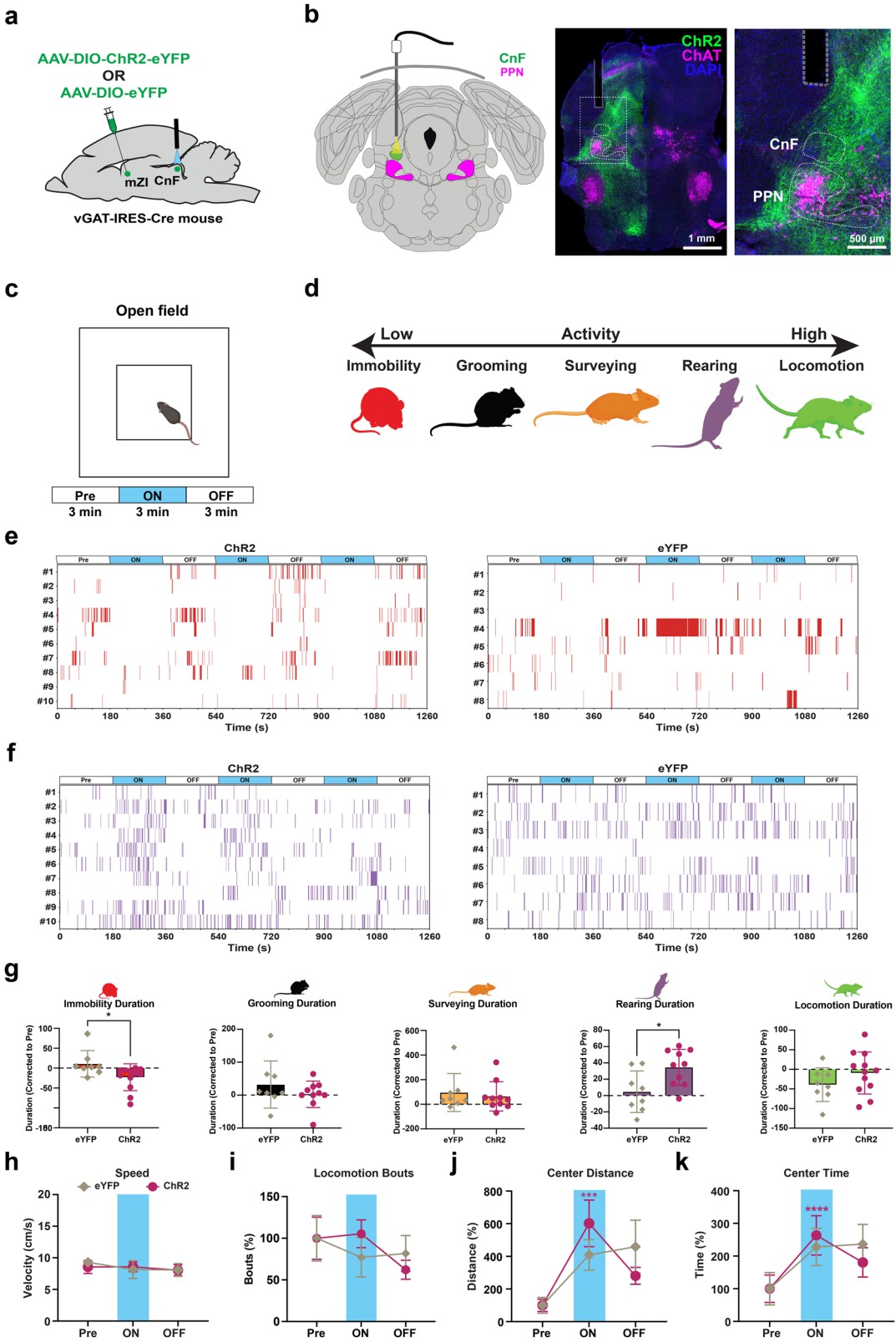

open field did not cause any change in the speed of locomotion, but did promote focused exploration (i.e. head-dips) in the hole-board arena. Although the CnF has been implicated in both walking and high-speed gaits, our data aligns with earlier studies showing that inhibition of CnF vGLUT2 neurons reduced the ability of mice to perform high-speed gaits like gallop and bound, without affecting slower speed gaits such as walk and trot[14]. Interestingly, chemogenetic inhibition of CnF

vGLUT2 neurons does not cause any change in spontaneous loco-motor speed during hole-board exploration[14]. In our work, mZI activity was inversely related to locomotor speed only under focused exploration in the hole-board. The inverse relationship found during focused exploration suggests a context-dependent control of speed, but we could not support this hypothesis with mZI stimulation. Alternatively, the increase in mZI activity is unrelated to the control of

**Fig. 6 | Photostimulation of mZI GABAergic terminals in the CnF increases open field exploratory behavior. a**, **b** Schematic depicts viral injection and photostimulation of mZI GABAergic terminals in the CnF; representative images show the optic fiber above mZI GABAergic terminals in the CnF (green) and PPN cholinergic neurons (magenta) ventral to the CnF. **c**, **d** Schematic illustrates open field locomotor activity testing protocol, with events classified as immobility (red), grooming (black), surveying (orange), rearing (purple), and locomotion (green). Ethograms showing immobility (**e**) and rearing (**f**) events during 3 min epochs of before (Pre), during (ON), and after (OFF) blue laser light (473 nm). **g** Panel shows total duration of behaviors during stimulation across trials after baseline correction to pre-stim conditions. Photostimulation of mZI GABAergic terminals in vGAT$^{ChR2}$ mice increased the proportion of immobility (ChR2: $n = 10$ mice; eYFP: $n = 8$ mice, $U = 22$, $p = 0.0469$, two-sided Mann–Whitney test) and rearing during laser ON

condition ($t_{(16)} = 2.68$, $p = 0.0163$, two-sided unpaired t-test). **h** Photostimulation of mZI GABAergic terminals in the CnF did not alter locomotion speed. **i** Total number of locomotion bouts were unchanged in vGAT$^{ChR2}$ mice compared to the vGAT$^{eYFP}$ mice. **j**, **k** An increase in the distance traveled in the center (ChR2: $n = 12$ mice; eYFP: $n = 8$ mice, $Q = 16.21$, $p = 0.0003$, Friedman test; *post hoc* vGAT$^{ChR2}$ Pre vs light ON, $Z = 3.98$, $p = 0.0002$; Dunn's multiple comparisons test) and time spent in the center of the open field (ChR2: $n = 12$ mice; eYFP: $n = 8$ mice, $Q = 17.91$, $p = 0.0001$, Friedman test; *post hoc* vGAT$^{ChR2}$ Pre vs light ON, $Z = 4.19$, $p < 0.0001$; Dunn's multiple comparisons test) was observed in vGAT$^{ChR2}$ mice in laser ON condition. Three trials per mouse were conducted and the data were normalized to baseline Pre condition within each group and presented as mean ± SEM. Atlas images adapted from the Allen Mouse Brain Atlas[68,69]. Illustrations in **a**–**d**, **g** created with BioRender.com. ***$p < 0.001$, ****$p < 0.0001$.

CnF mediated locomotion as argued earlier, but may contribute to maintenance of focused exploration. Indeed, we observed an increase in mZI activity during rearing and head-dipping. Activity of the mZI during different behaviors may reflect activity of different subpopulations which would not be resolved using fiber photometry. Our work does not exclude a role for the mZI in the control of speed, since it could potentially slow high-speed escape-related locomotion.

Although some reports suggest that head-dipping is associated with escape[49], photoactivation of CnF vGLUT2 neurons can produce escape behaviors during stimulation and suppression of head-dipping behavior, making this idea unlikely in the context of CnF activation[14]. The shift to defensive arousal with CnF stimulation has also been reported following activation of ventromedial hypothalamic glutamatergic neurons, evoking escape behavior[50]. It is likely that the increase in exploratory behavior during photoinhibition of CnF vGLUT2 neurons observed in our experiment is indicative of a behavioral switch promoting exploration. These behavioral switches are critical for the survival of animals while foraging in a naturalistic environment and/or facing the threat of predation.

Within the caudal mZI, vGAT neurons are active during hunting behavior and influence success rates when mice hunt for crickets[35]. Interestingly, they also report that mZI vGAT neurons are less active while eating prey or chow. Our data shows that mZI neurons show an step-wise increase in overall activity during focused exploration compared to thigmotactic locomotion, which is consistent with these reports. Also, the increase in exploratory behavior following terminal activation of the mZI-CnF GABAergic circuit is consistent with photostimulation of mZI GABAergic somata prompting appetitive hunting[35]. Activity of vGAT neurons in the ZI is increased following food restriction[27] and hunger is known to promote exploratory behavior and foraging[51]. Photostimulation of ZI GABAergic cells strongly inhibits active avoidance behaviors through projections to the PPN in mice. Inhibition of ZI GABAergic cells promotes active avoidance (footshock) behaviors, indicating the significance of context-dependency in this region[34]. Consistent with its role as a hub for sensorimotor integration[38], there are extensive projections to the ZI, which receives dominant inputs from bilateral cerebral cortex structures and the ipsilateral amygdala[52].

vGLUT2 neurons and likely vGAT neurons within the CnF receive mZI vGAT projections. vGAT neurons could contribute to the overall behavioral effect, which was more robust with terminal mZI stimulation. Evidence from our work does not support the activation of vGAT CnF neurons alone, as photostimulation of vGAT CnF neurons stops or reduces the speed of locomotion[14]. We did not observe either of these effects with mZI photostimulation, suggesting a more complex modulation of CnF populations. A recent study has elucidated the presence of subpopulations within excitatory neurons of the lateral rostral medulla of the brainstem, which have distinct connectomes and regulate distinct forelimb movements[53], and the same is true of the PPN[18]. It is likely that the CnF vGLUT2 neurons receiving mZI GABAergic projections have a distinct connectome and role in movement control.

One distinct PPN subpopulation with descending connectivity to the spinal cord can control vertical movement during rearing[18]. Interestingly, our photometry results also show that mZI GABAergic neurons are active during rearing and that photostimulation of mZI GABAergic terminals in the CnF can increase rearing. Our electrophysiology data show a depression in CnF neuron activity following mZI activation. While it is possible that this inhibition is indirect, the short latency of the depression is consistent with monosynaptic connectivity. This also agrees with photostimulation of the retrogradely labeled mZI-CnF circuit promoting exploratory behavior.

Although the optic probe was centered over the CnF, where functional connections were found, activation of mZI terminals could inhibit neighboring nuclei, such as the PAG or PPN. We tackled this issue with retrograde targeting of the mZI-CnF circuit which produced more targeted effects on hole-board exploration. This may suggest a more specific circuit from the mZI to the CnF targeting exploratory behavior. Taken together with the electrophysiology and anatomical data, this suggests that the mZI-CnF circuit is sufficient to produce exploratory behavior. The heterogeneity of PPN glutamatergic neurons is evident from one study showing promotion of exploratory behavior following activation of PPN vGLUT2 neurons[14], while another study shows promotion of avoidance response following activation of putative PPN glutamatergic neurons[54]. Given the heterogeneity of the PPN, we would expect more variability in our behavioral data, which were quite consistent. This suggests that mZI-PPN circuits do not significantly contribute to the effects reported here. Other work has shown that activation of mZI-PAG circuits can reduce defensive behaviors, while suppression of the circuit has the opposite effect[37]. The mZI projects to the PAG, specifically to excitatory neurons in the dlPAG and vlPAG[37]. The dlPAG has been shown to directly project to the CnF, triggering defensive locomotor behavior[43]. Thus, the mZI may promote a switch to exploratory behavior through direct and indirect inhibition of CnF vGLUT2 activity. While, as mentioned, mZI activation inhibits CnF neurons at short (<100 ms) latencies, an integrated circuit could augment the effects we observed.

Exploratory locomotion may be a consequence of reduced anxiety. Indeed, increased crossings into the center of the open field, increased time in the open arms of the elevated plus maze, and increased time spent in the light chamber of the light-dark transition test, are all possibly associated with anxiolytic effects. Similarly, open field thigmotactic locomotion has been reported when CnF vGLUT2 neurons are photostimulated[55]. Our findings and that of others raise an interesting possibility that the CnF, which is predominantly associated with motor function[16,18,55], can also affect emotionality. The presence of both ascending and descending outputs[23,56] of the CnF further suggest a multimodal role of CnF in motor and emotional behavior. When the retrogradely transfected GABAergic mZI-CnF neurons were photostimulated, we found the anxiolytic effects to be less profound. However, the increase in exploratory locomotor activity, as evidenced by the hole-board test, was retained. Overall, it suggests that some

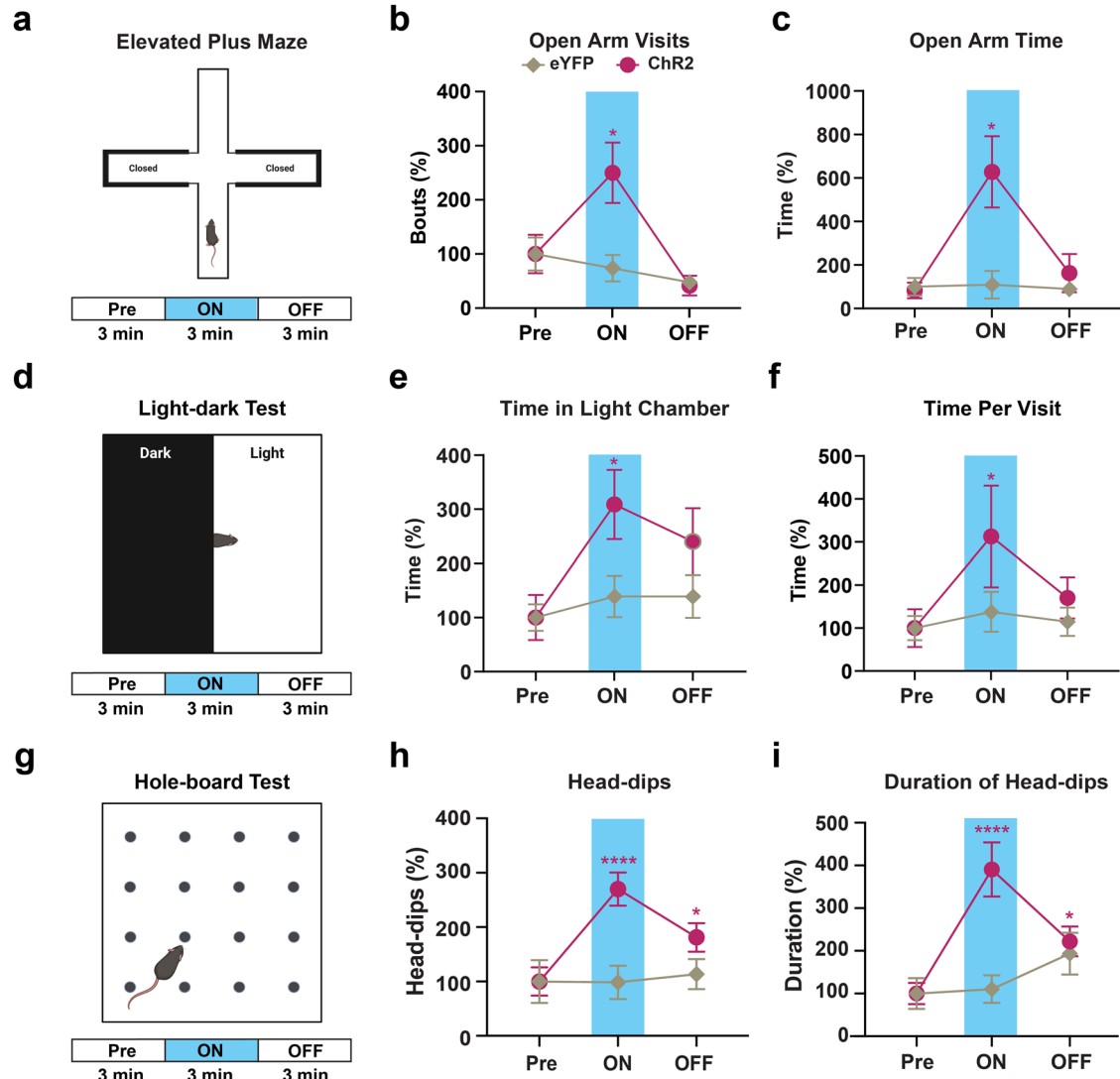

**Fig. 7 | Photostimulation of mZI GABAergic terminals in the CnF increases exploratory behavior. a–c** In the elevated plus maze photostimulation increased both the visits (**b**) (ChR2: *n* = 9 mice; eYFP: *n* = 5 mice, *Q* = 15.75, *p* < 0.0001, Friedman test; *post hoc* vGAT^CHR2 Pre vs light ON, *Z* = 2.83, *p* = 0.014, *post hoc* vGAT^CHR2 light ON vs light OFF, *Z* = 3.54, *p* = 0.0012; Dunn's multiple comparison test) and the time spent (**c**) in the open arms (*Q* = 11.70, *p* < 0.0013, Friedman test; *post hoc* vGAT^CHR2 Pre vs light ON, *Z* = 2.95, *p* = 0.0096, *post hoc* vGAT^CHR2 light ON vs light OFF, *Z* = 2.71, *p* = 0.02; Dunn's multiple comparison test). **d–f** In the light-dark transition test, photostimulation increased both the time spent (**e**) in the light chamber (ChR2: *n* = 9 mice; eYFP: *n* = 5 mice, *Q* = 8.3, *p* = 0.019, Friedman test; *post hoc* vGAT^CHR2 Pre vs light ON, *Z* = 2.83, *p* = 0.014; Dunn's multiple comparison test)

and the time spent per visit (**f**) to the light chamber (*Q* = 7.6, *p* = 0.02, Friedman test; *post hoc* vGAT^CHR2 Pre vs light ON, *Z* = 2.71, *p* = 0.020; Dunn's multiple comparison test). **g–i** In the hole-board test, photostimulation increased both the number of head-dips (ChR2: *n* = 17 mice; eYFP: *n* = 8 mice, *Q* = 22.88, *p* < 0.0001, Friedman test; *post hoc* vGAT^CHR2 Pre vs light ON, *Z* = 4.63, *p* < 0.0001; *post hoc* vGAT^CHR2 Pre vs light OFF, *Z* = 2.57, *p* = 0.030; Dunn's multiple comparison test) and their duration (*Q* = 18.47, *p* < 0.0001, Friedman test; *post hoc* vGAT^CHR2 Pre vs light ON, *Z* = 4.29, *p* < 0.0001; *post hoc* vGAT^CHR2 Pre vs light OFF, *Z* = 2.4, *p* = 0.049; Dunn's multiple comparison test). The data were normalized to baseline Pre condition within each group and are presented as mean ± SEM. Illustrations in **a**, **d**, **g** created with BioRender.com. *\*p* < 0.05, *\*\*p* < 0.01, *\*\*\*\*p* < 0.0001.

anxiolytic effects, as seen in the elevated plus maze, may have a threshold effect. This may be due to the possible differences in inherent anxiety effects for an elevated maze compared to other reported tests. In summary, multiple lines of evidence support the idea that the mZI-CnF inhibitory circuit promotes exploratory behavior, with some anxiolytic effects being modulated independently. Overall, the ZI is a complex region of the brain that is thought to play an important role in integrating motor and sensory functions and the control of behavior. The behaviors that it modulates include movement, emotion, arousal, and sleep. Moreover, the ZI is thought to contribute to rapid responses to changes in contextual demands, making it a critical coordinating hub. Our work illustrates that mZI GABAergic circuits to the CnF alter the behavior of mice by increasing the likelihood of exploratory behaviors.

## Methods
### Animals

vGLUT2-IRES-Cre (B6J.129S6(FVB)-*Slc17a6^{tm2(cre)Lowl}*/MwarJ; Cat. #028863) and vGAT-IRES-Cre mice (B6J.129S6(FVB)-*Slc32a1^{tm2(cre)Lowl}*/MwarJ); Cat. #028862) were purchased from The Jackson Laboratory (Bar Harbor, ME, USA). Mice were genotyped by the molecular core facility at Hotchkiss Brain Institute. Genotyping of the vGLUT2-IRES-Cre and vGAT-IRES-Cre mice was performed using the primers recommended by the supplier (The Jackson Laboratory, Bar Harbor, ME, USA). Mice were housed on a 12:12 h light:dark schedule (lights on 07:00 – off 19:00) at a room temperature of 20 °C and relative humidity of 34% with *ad libitum* access to food and water. Experiments were conducted in male mice between the ages of 8–16 weeks; previous work has shown no differences between male and female results in the

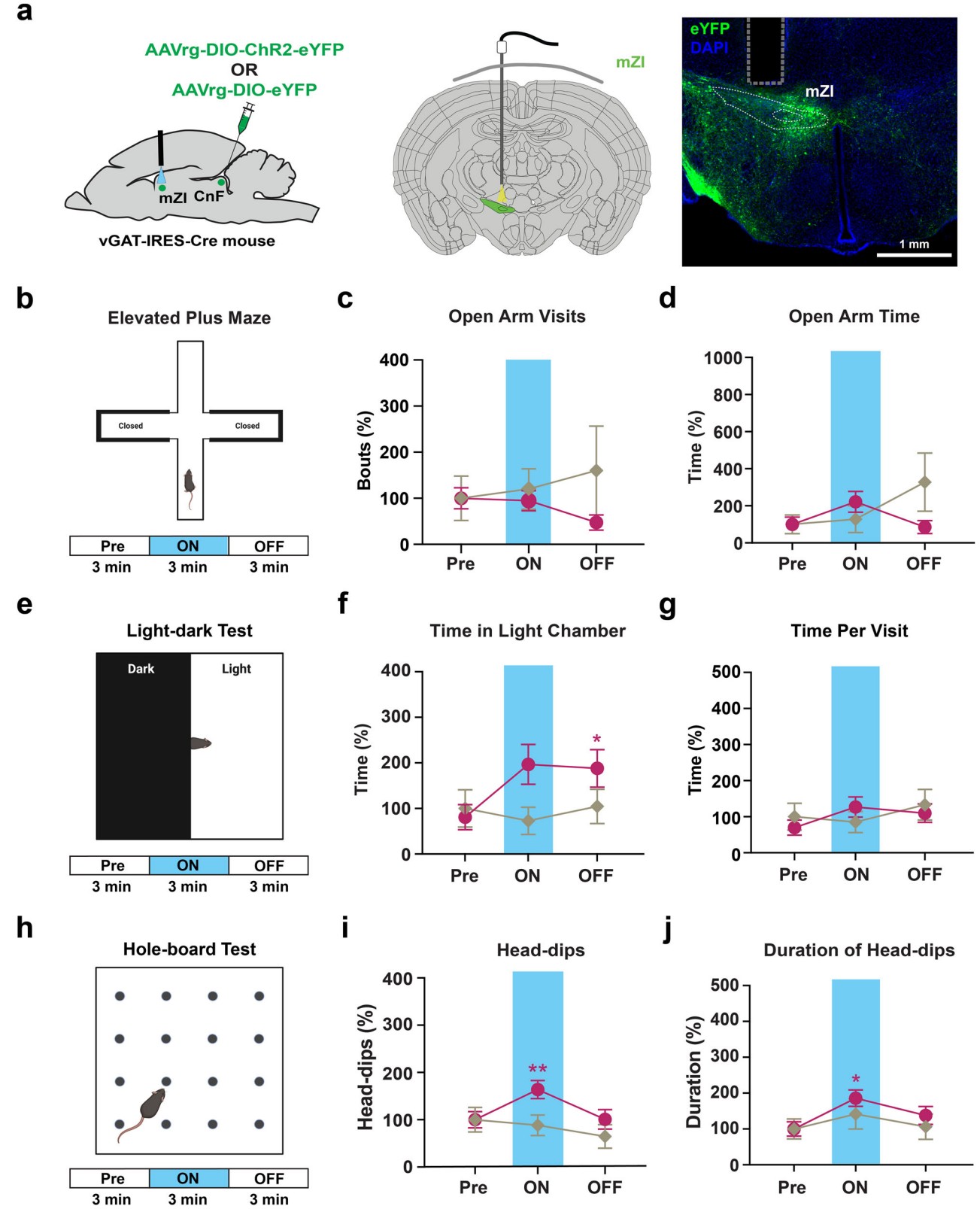

MLR. We have complied with all ethical regulations. All animal experiments were approved by the University of Calgary Health Sciences Animal Care Committee (Protocol #AC19-0035) and were conducted in accordance with the guidelines of the Canadian Council for Animal Care. Mice were housed in a double-barrier facility for breeding, in groups of 2–4.

**Viruses**

For optogenetic experiments, we used the following Cre-dependent recombinant AAVs: AAV9-EF1a-DIO-eNpHR3.0-eYFP (AAV-DIO-eNpHR3.0-eYFP, Addgene, Watertown, MA, USA, Cat. #26966-AAV9, Lot #v44904, titer 2.2 × 10^13 GC/mL), AAV9-EF1a-DIO-ChR2-eYFP (AAV-DIO-ChR2-eYFP, Addgene, Cat. #20298-AAV9, Lot #v45835, titer

**Fig. 8 | Photostimulation of CnF projecting mZI GABAergic soma increases exploratory behavior. a** Schematic shows viral injection and photostimulation of CnF-projecting mZI vGAT neurons; representative image displays ipsilateral mZI vGAT neurons (vGAT in green and DAPI in blue) projecting to the CnF and optic fiber tract location. In the elevated plus maze (**b**), no significant change in visits (**c**) to the open arms or time spent (**d**) in the open arms was observed (ChR2: *n* = 10 mice; eYFP: *n* = 6 mice). **e**–**g** In the light-dark transition test (**e**), an increase in the time spent in the light chamber occurred in vGAT$^{rgCHR2}$ mice in laser ON epoch (ChR2: *n* = 9 mice; eYFP: *n* = 6 mice, Time: $F_{(1.38,17.88)}$ = 4.64, *p* = 0.035; Time × Virus: $F_{(2,26)}$ = 7.26, *p* = 0.0031; *post hoc* vGAT$^{rgCHR2}$ Pre vs light ON, *p* = 0.025; *post hoc* vGAT$^{rgCHR2}$ Pre vs light OFF, *p* = 0.042, two-way repeated-measures ANOVA). Non-

significant increase in the time spent per visit (**g**) to the light chamber during blue laser stimulation of vGAT$^{rgCHR2}$ mice. **h**–**j** In the hole-board test arena (**h**), photostimulation increased both the head-dips (ChR2: *n* = 10 mice; eYFP: *n* = 6 mice, *Q* = 12.46, *p* = 0.00080, Friedman test; *post hoc* vGAT$^{rgCHR2}$ Pre vs light ON, *Z* = 3.019, *p* = 0.0076; *post hoc* vGAT$^{rgCHR2}$ light ON vs light OFF, *Z* = 3.019, p = 0.0076, Dunn's multiple comparison test) and their duration (ChR2: *n* = 10 mice; eYFP: *n* = 6 mice, Time: $F_{(1.49,20.79)}$ = 7.63, *p* = 0.006; *post hoc* vGAT$^{rgCHR2}$ Pre vs light ON, *p* = 0.0091; *post hoc* vGAT$^{rgCHR2}$ Pre vs light OFF, *p* = 0.020, two-way repeated-measures ANOVA). The data were normalized to baseline Pre condition within each group and are presented as mean ± SEM. Illustrations in **a**, **b**, **e**, **h** created with BioRender.com. \**p* < 0.05, \*\**p* < 0.01, \*\*\*\**p* < 0.0001.

1.8 × 10$^{13}$ GC/mL), AAV9-EF1a-DIO-eYFP (AAV-DIO-eYFP, Addgene, Cat. #27056-AAV9, Lot #v44841, titer 2.2 × 10$^{13}$ GC/mL), AAVrg-EF1a-DIO-hChR2(H134R)-EYFP-WPRE-HGHpA (AAVrg-DIO-ChR2-eYFP, Addgene, Cat. #20298-AAVrg, Lot #v32674, titer 1.0 × 10$^{13}$ GC/mL), and AAVrg-Ef1a-DIO-eYFP (AAVrg-DIO-eYFP, Addgene, Cat. #27056-AAVrg, Lot #v120252, titer 2.4 × 10$^{13}$ GC/mL. A Cre-dependent AAV construct containing GCaMP6s (AAV9.CAG.Flex.GCaMP6s.WPRE.SV40; Penn Vector Core, Philadelphia, PA, USA, Lot #CS1257, titer 4.6 × 10$^{13}$ GC/mL) was injected into the mZI of vGAT-IRES-Cre transgenic mice for the fiber photometry experiments. A Cre-dependent AAV construct (AAV-DJ-hSyn-Flex-mGFP-2A-Synaptophysin-mRUBY; Stanford Gene Vector and Virus Core, Stanford, CA, USA, Cat. #GVVC-AAV-100, Lot #5657, titer 4.7 × 10$^{12}$ GC/mL) was injected into the mZI of vGAT-IRES-Cre transgenic mice for anatomical tracing experiments.

### Stereotaxic surgery and optic fiber implantation
Mice were induced and maintained under isoflurane anesthesia (4% oxygen,1.5% isoflurane) in the stereotaxic apparatus (Stoelting Co., IL, USA). A glass capillary containing the viral vector was lowered into the brain (CnF**;** anterior–posterior (AP): −4.8 mm; lateral (L): −1.2 mm; dorsal–ventral (DV): −2.7 mm from the bregma) or (mZI**;** AP: −1.2 mm; L: −0.6 mm; DV: −4.5 mm from the bregma). A total viral volume of 50–100 nL was pressure injected with a Nanoject II apparatus (Drummond Scientific Company, Broomall, PA, USA). For optogenetic experiments, 200 μm wide mono fiber optic cannulas (Doric Lenses, Quebec City, QC, Canada, Cat. #MFC_200/240-0.22_3mm_FLT) were stereotactically implanted above the CnF for both eNpHR3.0 and ChR2 experiments (AP: −4.84 mm; L: 1.2 mm; DV: −2.6 mm from bregma). For optogenetic retrograde experiments, 200 μm wide mono fiber optic cannulas (Doric Lenses, Quebec City, QC, Canada, Cat. #MFC_200/240-0.22_5mm_FLT) were stereotactically implanted above the mZI (AP: −1.2 mm; L: −0.6 mm; DV: −4.5 mm from bregma). For photometry experiments, a 400 μm diameter mono fiber optic cannula (Doric Lenses, Cat. #MFC_400/430/0.48_5mm_MF2.5_FLT) was implanted dorsal to the mZI (AP: 1.2 mm; L: 0.6 mm; DV: 4.40 mm from bregma). Cannulas were affixed to the skull with Metabond® (C&B, Parkell, Brentwood, NY, USA) and dental cement (Dentsply, PA, USA). Mice were given 3 weeks to recover before the start of the experiment. *Post hoc* histology confirmed CnF$^{eNpHR3.0}$, mZI$^{ChR2}$, and mZI$^{GCamp6}$ transfection and optic fiber placement.

### Optogenetics
Following recovery, mice were handled on 3 consecutive days (5 min per day) and habituated to the fiber optic cable (Doric Lenses, Cat. # MFP_200/240/900-0.22_2.5m_FCM-MF2.5) attached to the mono fiber optic cannula in the testing room (15 min per day). Before any behavioral manipulation, mice were habituated to the experimental room 30 minutes before testing. For in vivo experiments, the laser light source (561 nm for eNpHR3.0: Laserglow Technologies, Toronto, ON, Canada, Cat. #LRS-0561) or (473 nm for ChR2: Laserglow Technologies, Cat. #LRS-0473-GFM) was controlled by transistor-transistor logic (TTL) pulses delivered using a Master 8 pulse stimulator (A.M.P.I., Jerusalem, Israel). For eNpHR3.0, yellow light

(561 nm at 5 mW) was used continuously for 3 minutes during each behavioral test. For ChR2, blue light was delivered continuously for 3 min during each behavioral test (473 nm, 20 Hz, 10 ms pulse width, 2 mW). For both ChR2 and eNPHR3.0 mice within the open field test, 2 to 3 epochs of laser onset alternating with laser off epochs were conducted.

### Electrophysiology
vGAT-IRES-Cre mice (*n* = 3) were injected with AAV-DIO-ChR2-eYFP. Mice were anesthetized (isoflurane 5% induction, 1.5% maintenance, in 1 L/min O$_2$), and a mono fiber optic cannula was implanted directed at the mZI (AP: −1.4 mm, L: 0.8 mm, DV: −4.0 mm) and secured with Metabond® Quick Adhesive Cement System and dental cement. A single-wire electrophysiology probe (25 μm tungsten, California Fine Wire, Grover Beach, CA) was inserted above the CnF (AP: -4.8 mm, L: +1.0 mm from the bregma) and advanced ventrally by 50–100 μm steps to record individual cells (MCU, Spike Gadgets, San Francisco, CA, USA). When putative spiking was detected, cell firing was recorded for a 1-minute baseline, followed by 6 single optogenetic pulses (10 mW, 10 ms pulse duration, 5 s ITI) and 6 burst stimulations (5 pulses, 10 Hz, 10 ms pulse duration, 5 s ITI). In 1 session, serial recordings were obtained from 20 to 40 cells. Putative spikes were sorted based on standard electrophysiological characteristics (Plexon offline sorter). Recordings with unstable pre-stimulus baselines (firing rate changes greater than 5x across trials) were discarded. Firing rate changes, relative to the pre-stimulus baseline, were calculated in response to burst stimulation for each trial in each recording to calculate a one-sample t-test for each cell. A false discovery rate correction was used to adjust the *p* values calculated for each cell[57]. From a total of 69 cells, 27 were histologically determined to be within CnF. The stereotaxic location of each recording site relative to the surface of the brain was recorded during the experiment. To verify this estimate *post hoc*, the deepest location of each recording tract was lesioned, and the location of each recording was revised relative to this histologically verified final location of the probe[58]. Based on the statistical criteria, 7 of 27 cells were categorized as responding cells, with all responding cells decreasing firing rates in response to optogenetic stimulation.

### Fiber photometry
We recorded Ca$^{2+}$ transients from mZI vGAT neurons of freely moving mice using fiber photometry. Animals were handled for 15 min on 3 successive days, and then habituated to the fiber optic cable in their homecage (30 min per day). We recorded 5 minutes of mZI vGAT neuronal activity in the homecage immediately before each 10 min test. A Doric color camera (Doric Lenses, Cat. #BTC_USB3.0_CO) was fixed in place above the arena and synchronized. A Doric fiber photometry system, consisting of two excitation LEDs (465 nm and 405 nm from Doric Lenses) was controlled by an LED driver and console running Doric Neuroscience Studio software (V5, Doric Lenses). The LEDs were modulated and the resulting signal demodulated using lock-in amplification. Both LEDs were connected to a Doric Mini Cube filter set (Doric Lenses, Cat. #FMC4_AE(405)_E(460-490)_F(500-550)_S) and the

excitation light was directed to the animal via a mono fiber optical patch cord (Doric Lenses, Cat. #MFP_400/460/1100-0.48_2m_FC/MF2.5). The power of the LEDs was adjusted to provide 30 μW at the end of the patch cord. The resulting signal was detected by a Femtowatt Silicon Photoreceiver (Newport Corp., Irvine, CA, USA, Cat. #2151). After each recording, signal and reference data were decimated by a factor of 100. The data were analyzed using a custom-adapted MATLAB (R2022, MathWorks Inc., Natick, MA, USA) script[59–61]. Briefly, changes in fluorescence intensity were recorded by sampling the 465 nm wavelength, while the isosbestic 405 nm wavelength was used to control for movement artifacts. Signals were first resampled to a rate of 100 Hz. Artifacts were removed and replaced with a linear interpolation by user-specified regions. The data were low pass filtered (2 Hz) and fitted to an exponential decay to correct for photobleaching. After using a polynomial fit function, movement artifacts were removed by subtracting the 405 nm signal from the 465 nm signal. Data was normalized using the function $(f - f0)/f1$ where $f0$ represents the mean and $f1$ represents the standard deviation within a moving window of 60 s. The area under the curve was calculated as the normalized averaged z-score between the start and stop condition epochs specified through tracking software (TopScan Version 3.0, Clever-Sys Inc., Reston, VA, USA) or manual annotation through an event logging software called Behavioral Observation Research Interactive Software (BORIS: Version 8.2)[62]. Initiation triggered values were calculated by averaging z-score traces 5 s before and after behavior initiation and further averaging across all mice. Baseline values were calculated by averaging values between −5 and −3 from the averaged z-score traces. Corrected max/min was computed by taking either the maximum or minimum between 0 to 3 s and subtracting the baseline value. Locomotor bouts were correlated to z-score values by extracting timestamps from the photometry videos through a custom script in MATLAB (MathWorks Inc.).

## Behavioral tests

Each apparatus was cleaned with a 70% ethanol solution to eliminate residual odors from other mice. Behavior was recorded using a vertically mounted video camera and analyzed with the TopScan video tracking software (Clever Sys Inc.). BORIS[62] was used offline by an experimenter who was blinded to the treatment conditions to manually score behavioral events such as rearing, head-dipping, etc.

## Open field

Behavioral testing was performed in a 50 × 50 × 50 cm open field chamber. Each mouse was habituated to the chamber for 3 days for 30 min per day before testing. On test days, mice were introduced to the chamber for testing and trials were performed in ABA fashion with 3 min epochs of laser light ON or OFF during a test session. We performed 2-3 repetitions of laser ON and OFF epochs and averaged the results. We used the manual behavioral annotation software, BORIS[62], and the Clever Sys video tracking software, TopScan (Clever Sys Inc.), to divide movement into the following categories:

(1) Immobility: defined by both speed and motion measures (animal body change in consecutive frames expressed as a percentage). Immobility bouts over 15 successive frames were estimated when speed was less than 6 cm/s and the change in motion was less than 1% change in animal body position.
(2) Grooming: a stereotypical behavior that involves self grooming with forepaws and licking of fur, usually in a sitting position.
(3) Rearing: mice put their weight on hind legs, raise their forelimbs from the ground, and extend their head upwards.
(4) Surveying: occurs when the animal stays in one place with some observed movement such as sniffing, stretching, turning around, and circling. However, the animal is neither immobile nor locomoting.

(5) Locomotion: characterized by the distance traveled, locomotor bouts, and locomotor duration. Distance traveled: the mouse physically moves or displaces itself from one location to another with a minimum prescribed speed greater than 60 mm/s and a minimum distance of 100 mm. Locomotor bouts: the number of times an animal moves from one place to another place. If the animal continuously moves, it is counted as one single bout. Locomotor duration: the amount of time the mouse spends in locomotion.

## Elevated plus maze

The elevated plus maze is a 'plus'-shaped maze (CSM Opto facility) comprised of two facing open arms (8.5 × 21.5 cm) and two closed arms (8.5 × 21.5 × 22 cm), 51.5 cm above the floor. During optogenetic stimulation, mice were tested for a total of 9 min in 3 min epochs (in ABA fashion) under bright white light. The time spent in the open arms, central junction, and closed arms, as well as the number of entrances and explorations in each section, were recorded using a video camera and video tracking software (TopScan, Clever Sys Inc.). Area under the curve (AUC) analysis was performed using custom scripts (MATLAB, MathWorks Inc.).

## Light-dark transition test

Mice were tested for natural aversion to bright illumination and spontaneous exploratory behavior during light-dark tests. The box consisted of one dark chamber and one brightly lit transparent chamber connected with a doorway in the center, allowing the transition from one compartment to another. Mice were introduced into the dark chamber, and an ABA design of 3 min epochs was followed for a total test duration of 9 min. Transitions were recorded using a video camera and video tracking software (TopScan, Clever Sys Inc.); the duration and frequency of transitions to the light chamber were calculated.

## Hole-board

Behavioral testing was performed in a 45.5 × 45.5 × 45.5 cm chamber with 16 equidistant holes (2 cm diameter). Mice were habituated to the testing room 30 min per day over 3 days before the experiment. During habituation, the patch cable was connected to the optic fiber in the homecage. On testing day, after mice were acclimated for 30 min, baseline photometry measurements were recorded while mice were in the homecage. After 5 min, mice were placed in the hole-board arena for 10 more minutes. Head-dip, rearing, and immobility behaviors were manually annotated through BORIS[62]. Head-dip behavior consisted of the time after the mouse approached a hole and fully submerged its head into the hole leading up to the time it would retreat. Immobility consisted of periods where the mouse would cease any movement. Rearing was defined as periods where the mouse would extend both forepaws vertically either in the air or resting on a wall. Grooming was excluded from the rearing definition. Thigmotactic locomotion was scored using video tracking software (TopScan, Clever Sys Inc.), where thigmotactic zones were defined as the periphery (25%) of the hole-board arena. Focused exploration in comparison to thigmotactic locomotion in the context of velocity was quantified using custom scripts (MATLAB, MathWorks Inc.) and is described in the following passage. The mouse center-of-mass location within the hole-board was tracked using custom algorithms (MATLAB, MathWorks Inc.). Instantaneous velocity at each frame was calculated using gaussian smoothed position (1 s window). Pauses were defined as velocity less than 3 cm/s continuously for at least 1 s and locomotion initiation bouts were aligned at the termination of the pause. Focused exploration epochs were defined as two or more successive head-dips within 5 s, beginning at the first head-dip and ending at the last head-dip. Focused exploration episodes constituted 5 to 15% of total time for all mice.

## Immunohistochemistry

Mice were deeply anesthetized with isoflurane (2%) and transcardially perfused with phosphate-buffered saline (PBS), followed by 10% neutral buffered formalin (NBF). The animals were decapitated, and the whole heads were incubated overnight in 10% NBF at 4 °C before the fiber optic was removed and the brain was extracted from the skull. Brains were incubated in 10% NBF for up to 4 h followed by 30% sucrose (w/v in PBS) for cryoprotection. 50 μm coronal brain sections were obtained using a cryostat (Leica Biosystems, ON, CA, Cat. #CM1850 UV). The sliced brain sections were collected in a staggered fashion and placed into 3 consecutive wells. Sections were rinsed before and between incubations with 0.1 M PBS. Sections were incubated in blocking solution (5% donkey serum in PBS with 0.5% Triton-X 100 (PBST), Sigma-Aldrich, St. Louis, MO, USA) for 1 h and blocking solution (5% donkey serum in 0.3% PBST) was used in subsequent antibody incubations. Controls were included where the primary antibody was omitted to check for non-specific binding of the secondary antibodies. Free-floating brain sections were then mounted onto Superfrost™ slides in VECTASHIELD® PLUS Antifade Mounting Medium (Vector Laboratories Inc., Newark, CA, USA, Cat. #VECTH190010) and cover-slipped. All primary and secondary antibodies, along with their conjugates, are presented in Supplementary Table 1.

## RNAscope®

vGAT-IRES-Cre mice were transfected with a Cre-dependent AAV construct (AAV-DJ-hSyn-Flex-mGFP-2A-Synaptophysin-mRUBY) in the mZI and transcardially perfused with chilled 1X PBS followed by 10% NBF. Brains were extracted and post-fixed in 10% NBF for 24 h at 4 °C and then immersed in 10%, 20%, and 30% sucrose in PBS on subsequent days. The tissue was embedded in an optimal cutting temperature (OCT) compound, sliced at 12 μm thickness, mounted on Superfrost Plus™ slides, and stored at −80 °C. The RNAscope® fluorescent multiplex kit was used (Advanced Cell Diagnostics (ACD), Newark, CA, USA, Cat. # A320850). Slides were washed with 1X PBS for 5 min to remove OCT and then baked in the HybEZ™ II Oven (ACD, Cat. #321721) for 45 min at 60 °C. Slides were post-fixed by immersion in chilled 10% NBF for 30 min at 4 °C. Slides were then dehydrated in a graded ethanol series (50%, 70% and 2 ×100% ethanol) for 5 minutes and then left to air dry for 5 min. Slides were placed into boiling 1X Target Retrieval solution (ACD, Cat. #322000) for 6 min and immediately washed in distilled water and 100% ethanol. A hydrophobic barrier was drawn (Vector Laboratories Inc., Cat. #H-4000) around the tissue and Protease III (ACD, Cat. #322340) was applied to each slide and baked for 30 min at 40 °C. Slides were washed with distilled water, then warmed RNAscope® probes were mixed, added to the slides, and incubated for 2 h at 40 °C. An antisense probe for vGLUT2 was used (ACD, Cat. #319171-C3). After washing the slides twice with 1X Wash Buffer (ACD, Cat. #310091) for 2 min each, we followed the manufacturer's recommendation to amplify the in situ hybridization signal. Slides were counterstained with DAPI for 30 s then mounted with ProLong™ Glass Antifade mountant (Invitrogen, Waltham, Cat. #P36984) and coverslipped.

## Microscopy and image analysis

Immunohistochemistry fluorescent images were collected with the Leica TCS SP8 microscope and Olympus VS110 Slide Scanner microscope. Offline image processing included maximal intensity projections and mean fluorescence intensity (Supplementary Fig. 2b) was conducted using Fiji[63]. RNAscope® images were captured with the Nikon A1R-MP multiphoton confocal microscope at 20X magnification and 2.25X optical zoom. RNAscope® images captured nuclear stain (DAPI), SYP-mRUBY[+] and RNAscope® probe signal (vGLUT2). The background was subtracted with a rolling ball radius of 20 pixels in Fiji (version 1.53f51)[63]. Nuclear staining from RNAscope® images were imported into Ilastik (version 1.3.3 post3)[64], a machine-learning software for bioimage analysis, where a pixel classification model was trained and binary images were re-merged with the original signal channels. Images were then imported into QuPath (version 0.2.3), an open-source software for bioimage analysis[65]. Cell detection and subcellular spot detection functions were performed on the images to obtain detection measurements of each nuclei. To validate the accuracy of Ilastik pixel classification, we manually counted the DAPI signal in 26% of all analyzed images (18 out of 68 images). Manual counts were compared to the total cell count values exported from QuPath for the corresponding images. We found that QuPath was overcounting nuclei by an average of 4.33 (±3.99) cells, which is a 2.63% (±2.44) overestimation of cells ($M = 164.56 \pm 7.37$ cells per image). The detection measurements from each image were analyzed using a custom script in MATLAB (Mathworks Inc. 2021b) to sort and calculate colocalization for SYP-mRUBY[+] and vGLUT2[+] puncta. We considered cells to be SYP-mRUBY[+] or vGLUT2[+] if they had ≥4 puncta associated with the nucleus ROI[66]. Images were also collected using the Olympus VS110 Slide Scanner microscope to visualize the extent of SYP-mRUBY[+] puncta labeling. To separate puncta from intensely labeled vasculature, we performed post-processing using IMARIS (version 9.9.1, Oxford Instruments, Abington, UK). We applied a spot detection function with a diameter of 2.5 μm with background subtraction, and then adjusted the quality filter threshold to detect the majority of puncta. We used a machine learning model to classify the puncta, generated a mask from the classified puncta, and exported it to Fiji[63].

## Data analysis & statistics

All data were tested for normality using the Shapiro–Wilk normality test. Data are reported as mean ± SEM. Data that were normally distributed were subsequently analyzed using parametric tests. Analysis of behavioral data were performed using repeated-measures one or two-way ANOVAs with multiple comparisons or a mixed-effects analysis were performed. Bonferroni and Tukey post hoc tests were used to determine significant pairwise comparisons. When assumptions of normality were violated on the ANOVA, non-parametric Friedman tests were performed with Dunn's multiple comparison tests when significant main effects were found. Unpaired Student's $t$ tests were conducted comparing two independent groups, and a Mann–Whitney test was performed when assumptions of normality were violated. Prism 10 software (GraphPad Software, San Diego, CA, USA) was used for all statistical analyses, and an alpha of $p < 0.05$ was used.

## Reporting summary

Further information on research design is available in the Nature Portfolio Reporting Summary linked to this article.

# Data availability

The data generated in this study have been deposited in the Open Science Framework (OSF) database[67]. Databases used in this study: Allen Reference Atlas[68]. Source data are provided with this paper.

# Code availability

Data were analyzed using custom scripts. Scripts are available in the Supplementary Software ZIP file. All scripts are also available on OSF[67].

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

## Acknowledgements

We acknowledge grant support from the Canadian Institutes of Health Research (PJT-173511) and support from the Frank Leblanc Chair in Spinal Cord Injury Research. We also acknowledge technical assistance from Mr. Leo Molina for custom MATLAB scripts used in fiber photometry analysis. Many thanks to Joseph Grams for editing the final document. Furthermore, we acknowledge technical support from the HBI-AMP core microscopy and the CSM-Opto behavioral facility at the University of Calgary.

## Author contributions

SS performed experiments and prepared figures. SS, MAT and PJW edited figures. SS and PJW conceived and designed the research and interpreted results. PJW procured funding for the experiments; AM procured funding for the ePhys equipment and supplies. SS performed surgeries for tracing, optogenetics, fiber photometry and Opto-ePhys experiments. SS conducted behavioral experiments for optogenetics, analyzed data and prepared figures. CAB conducted behavioral experiments for fiber photometry, CAB, DMA and SS analyzed the data and prepared the figure. DMA performed the Opto-ePhys experiment, analyzed data and prepared the figure. The ePhys work was performed in the lab of AM. AM interpreted the results, contributed to research design, and edited the figures and manuscript. SAD performed an RNAscope experiment and quantification; SS and SAD analyzed data and prepared the figure. SS, MAT, CAB, GKG and CRB performed immunohistochemistry. SS, MAT, CAB, SAD and GKG performed imaging. SS, CAB, ZG, GKG, MAT, CRB and SAD manually analyzed hole-board data in BORIS. SS and PJW drafted the manuscript. All authors reviewed and approved the final version of the manuscript.

## Competing interests

The authors declare no competing interests.
