## [Peer Review File · Nature Communications]

Inhibitory medial zona incerta pathway drives exploratory behavior by inhibiting glutamatergic cuneiform neuronsREVIEWER COMMENTS

Reviewer #1 (Remarks to the Author):

In this manuscript, Sharma et al. examine the role of the inhibitory medial zona incerta (miZI) pathway to the cuneiform nucleus (CnF) in exploratory behavior. Using a combination of optogenetic manipulation, viral tracing experiments, and calcium fiberphotometry in freely behaving transgenic mice, they conclude that GABAergic miZI neurons initiate exploratory behavior by inhibiting glutamatergic CnF neurons.

This is a novel pathway and a nice combination of genetic techniques, but the current version of the paper is missing some detail and information about the experiments, lacks a clear definition of foraging behaviors, and does not sufficiently describe neural mechanisms. More importantly, discrepancies in the results disagree with the initial interpretation of data and in contrast suggest that the CnF-miZI pathway would be unlikely involved in foraging and exploration.

Major concerns:

1. The paper in its current format looks like a short communication without clearly identified sections (introduction, results, and discussion). Given the expertise of the lab in the ZI (Koblinger et al.), it would be important to introduce more background regarding the nature of ZI neurons and their projections in the introduction and discussion. Also, given the importance of behavioral experiments and the small number of figures in the current manuscript, extended data 1, 2, and 3 could be included as figures in the paper.

2. Behavioral analysis in extended data 1 and 3 is insufficient in the current presentation. Given that statistical analyses of percentages between stimulated versus control animals (S1e versus S1f) will unlikely be significant, the authors should statistically analyze whether photoinhibition of glutamatergic CNF neurons and photostimulation of GABAergic miZI neurons induce changes in the proportion of behaviors in comparison to the prestimulation period for each mouse. The absence of color (white strips) should also be explained. The definition of "in place activity" is too broad and misleading as it represents a combination of different behaviors including grooming, sniffing, stretching, and rearing, of which some are already identified as a class (at least for grooming and rearing). A similar analysis should also be included to study foraging locomotion and locomotion in the hole-board test in Fig. 3.

3. Figure 1 describes behavior during the open field, elevated plus maze, and light-dark tests upon photoinhibition of glutamatergic neurons of the CnF. The absence of change in locomotor speed upon photoinhibition of glutamatergic neurons of the cuneiform is really surprising and should be carefully explained and discussed in regard to previous genetic studies reporting a decrease in step cycle duration and locomotor speed upon inhibition of glutamatergic neurons of the cuneiform (Caggiano et al., 2018; Josset et al., 2018). The increased number of locomotor bouts in absence of changes in speed might suggest that animals were doing more pauses and more short, faster locomotor bouts. It will therefore be important to calculate the locomotor speed for each locomotor bout to exclude this possibility. Furthermore, the absence of significant differences in bouts of locomotion (Fig. 1f), rearing (Fig. 1i), and time in light chamber (1n) appear really unexpected in regard to the SD. Please double-check the statistics!

4. Retrograde tracing experiments in Fig. 2 illustrate low and intermediate magnification images of back-labeled GABAergic miZI neurons projecting into the CnF. The quality of images using multiphoton microscopy is great, but it would be useful to quantify the number or density of back-labeled neurons and report the nature of their projections. Are they bilateral or unilateral? The schematic in the extended data S4 gives the impression that the miZI would project contralaterally to the CnF. Is that correct? Furthermore, the retrograde serotype AAVr-ChR2 construct could have been used in functional studies to assess the role of miZI neurons projecting to the CnF and confirm whether their direct activation modulates locomotor behavior. This photoactivation experiment of miZI neurons would allow the authors to refine the results generated in Figure 4, using direct photoactivation of GABAergic miZI terminals to the CnF, which is not really specific as it labeled neighboring structures such as the PAG and the PPTg (see also my comments in the next section).

5. The combination of anterograde tracing with RNAscope in Fig. 2 to identify the phenotype of CnF neurons is an elegant approach, but the extent of labeling raises some concerns as it is not circumscribed to the CnF; it also projects to the dorsolateral PAG and the lower part of the SCP, suggesting that it might also contact parts of the PPTg. Given the role of these last nuclei in locomotor behavior and exploratory behavior for the PPTg (Caggiano et al., 2018), it will be important to quantify and illustrate the extent of these terminals at low magnification at the brain level and at high magnification at the nuclei level.

6. The density of synaptic puncta of VGAT miZI neurons onto VGLUT2 CNF neurons should be quantified. The authors should discuss their results regarding the presence of 40% of non VGLUT2 (presumably GABAergic) CNF neurons that receive GABAergic miZI inputs. Could it be a disinhibitory miZI-CNF pathway? If so, that's a novelty and it should be mentioned.

The legend of Fig. 2e-f should be corrected. It should be clarified whether images were collected using multiphoton or confocal microscopy in legends and methods. The use of machine learning is interesting, but it is not clear how it was used. Was there any validation of the predictions?

7. The definition of foraging locomotion in Fig. 3 is problematic. How can animals forage and walk at the same time? One can expect animals to move around their food or an object while foraging or exploring, but can we really call that locomotion? Using the behavioral classification (extended data 1 and 3), the authors should be able to more specifically distinguish foraging behaviors from locomotion.

8. There are some discrepancies in Fig. 3 and S2. How can calcium activity of GABAergic miZI neurons be increased immediately prior to initiation of foraging locomotion in Fig. 3, but also be increased about 1 second after the initiation of head-dip in Fig. S2? In all cases, profiles of calcium transients in Fig. 3 should be analyzed for all behavioral conditions during the hole-board as well as in the open field. Calcium transients should also be analyzed as function of the locomotor speed.

Concerning potential technical biases, would it be possible to confirm that calcium activity is unrelated to the animal's head movement while head-dipping in the hole?

Furthermore, although the hole-board test is used as an exploratory test, it has also been considered as an escape behavior in some studies (Brown and Nemes, 2008)! Perhaps this should be mentioned and discussed.

9. In absence of changes in locomotor speed (Fig. 4e) and bouts (4f), the increase in distance (4g) and time (4h) travelled in the center of the open field as well as the number of rearings (4i) upon PS of GABAergic miZI terminals in the CNF is a bit surprising and unexpected. Can the authors explain? Moreover, the large SD in Fig. 4h suggests that the time spent in the center of the open field is unlikely significant. Please double-check the statistics.

10. I have some difficulties in reconciling the increased calcium activity (Fig. 3g) of GABAergic miZI neurons during the "pause" prior to initiation of "foraging locomotion" (Fig. 3f) and the absence of change in locomotor speed (Fig. 1e and 4e) in response to optogenetic inhibition of glutamatergic CNF neurons or optogenetic activation of GABAergic miZI terminals in the CNF. This is inconsistent and suggests otherwise that the CnF-miZI pathway would be unlikely involved in foraging or exploration.

Minor comments:

Legend of Figure 2:

2d: Rephrase: presence of synaptic puncta (in magenta) of miZI inhibitory fibers "in the CnF" innervating CnF glutamatergic vGLUT2 neurons (in green) showing presence of vGLUT2 mRNA.

2e: Cells in the CnF cannot express but colocalize with SYP-mRUBY synaptic puncta of GABAergic miZI terminals.

The label and legend in Fig. 2e are not clear: What does the author mean by VGLUT2 neuronal projections in VGAT-CRE mice?

Statistics: I assume that the stats represent the mean +/- SD. Please mention this in the figures

or methods.

Page 2: Recent studies have shown (only one study is referenced).

Page 3: VGAT2ChR2 should be VAGTChR2

Reviewer #2 (Remarks to the Author):

This is an interesting study. In this study, they found a new ZI-CNF circuitry for claiming the role in exploratory behaviors. However, it seems that their experimental results cannot clearly support their claims in the current version of the manuscript. I think that the authors need substantial revisions with additional experiments to clarify their claims.

First, the definition of exploratory behaviors they suggested is not clear. They described "These results provide evidence for context-dependent dynamic regulation of CnF vGLUT2 neurons, with the mZI-CnF circuit providing an important contribution in foraging and exploratory behavior." (page 1). For examining the causal role of ZI-CNF circuitry for exploratory behaviors, they did open-field test, elevated plus maze and light/dark test with optogenetic excitation/inhibition experiments. In the previous many papers, open-field (calculating the duration in center zone), elevated plus maze and light/dark test have been used for claiming anxiety. Even though the complete, conceptual separation of exploration and anxiety is quite difficult in experimental conditions, they need to sophisticatedly perform more additional experiments for explorative behaviors like hole-board test (the authors already did for fiber-photometry in Fig.3) or other types of exploratory behaviors with optogenetic excitation/inhibitions.

Second, they did hole board test only for fiber-photometry. There are logical gaps in the experimental design and the interpretation of the role of ZI-CNF circuitry in anxiety-like behaviors (Fig. 1 & 4) or hole board explorative behaviors (Fig. 3). The authors need to clarify the mechanistic link between the results from Fig1, Fig4 and from Fig.3.

Third, in the paper of Caggiano et al., (2018, Nature, <https://www.nature.com/articles/nature25448>), they show the chemogenetic inhibition of CNF_vglut2 neurons does not affect the exploration with hole board tests. How do the authors interpret these results considering the author results?

Reviewer #3 (Remarks to the Author):

In the manuscript "Inhibitory medial zona incerta pathway drives exploratory behavior by inhibiting glutamatergic cuneiform neurons", the authors show that inhibition of glutamatergic neurons in the cuneiform nucleus (CnF) leads mice to spend more time in the center of an open field box and in the open arms of an elevated plus maze. They found that GABAergic neurons from the mZI project to the CnF and are active during exploratory behavior, in particular head-dipping during hole exploration. Activation of the mZI to CnF projection results in similar exploratory behavior as inhibition of the glutamatergic CnF neurons. The manuscript concludes that this inhibitory projection promotes exploratory behavior. This adds an interesting new finding to recent work on the roles of the CnF and the ZI in motivated behaviors.

This conclusion is supported by the data and previous literature. In particular because locomotion by itself does not seem strongly affected by the optogenetic manipulations, although the manuscript is a little unclear about this point. I believe that the experiments by themselves also allow the conclusion that the primary action is a reduction in anxiety, indirectly leading to an increase of exploratory behavior. Previous literature on the CnF and mZI, however, support the conclusion reached in the manuscript that the projection drives an increase in exploration, independent of anxiolytic effects.

One surprising omission in the presented experiments is the combination of the hole-board test with activation and/or inactivation of the mZI to CnF projection. This would show more clearly the exploratory nature of the behavior. The fiber photometry for the hole-board test has been done for the mZI neurons, and not for the CnF neurons or the mZI to CnF projection (if the latter is possible). The tests presented for the mZI to CnF are a little less specific for exploratory behavior. I am reluctant to ask to this whole new set of surgeries and experiment, but it would make the evidence for the conclusion much stronger.

Besides this point, I only have a number of small comments (roughly in order in which they appear in the manuscript).

In Fig 1b-c, the dorsal atlas outline does not fit the slice at all, and with the image quality of the review manuscript, it is difficult to judge whether the real slice is close to the -4.84 mm atlas slice that the projected CnF outline is close to the real outline of the CnF. Is it perhaps possible to adjust the image settings or upload a better quality picture to allow the reviewer, and the reader, to better judge the accuracy of the fiber placement?

Fig 1b-c. Please add which atlas outline is used.

Fig 1f. The corresponding caption and text suggests significant change in number of locomotion bouts, but no star is shown. Is this right?

Fig 1i. The corresponding caption suggests significant change in number of rearings, but no star is shown. Is this right?

Fig. 1. Is the number of animals written somewhere? I cannot find it. If not, please add number of animals to captions, corresponding text or elsewhere.

Fig. 1n. No star is shown. Is this correct?

"Our results that show the inhibition of CnF vGLUT2 neurons increasing exploratory behavior proposes the CnF as a critical part of the defensive behavior circuit" This seems too strong a conclusion at this point, because inhibition of these neurons has shown effects that could also still be interpreted as a drop in anxiety. I would suggest phrasing it less strongly.

Fig. 2a. Please mention how the ZI outline was determined and drawn.

"Consistent with other reports". Please add references to these reports.

Fig. 2c. What is the relationship between the rectangular region outlined in the left panel and the square images on the right?

Fig. 2d. What do the arrow heads mean?

"Our results show that CnF cells are predominantly glutamatergic (Figure 2e: Mean = 65.9 ± 5.21 ; $p = 0.0403$)," What does this p-value mean? What is tested? Is this a useful test? What are the dots, slices or mice?

"nearly half of the vGLUT+ cells were associated with SYP-mRUBY synaptic puncta" Indeed, and given the larger number of VGLUT+ cells, I am very willing to believe that they are recipient of the input from the mZI, but the image Fig 2d does not really show very well that all these associated SYP-mRUBY puncta indeed contact the VGLUT+ positive cells. The word "confirm" in the line "These results confirm the presence of a mZI-CnF circuit providing inhibitory input to the excitatory vGLUT2+ cells" is a bit strong, something like "strongly suggest" is more accurate, I feel. Patching vglut2-cells and (optogenetic) activation of mZI cells/projections would be the way to really confirm this.

"(Figure 2e: Mean = 49.15 ± 51.6 ; $p = 0.8774$)." Is this a useful test?

Fig. 3 e . Is a df/F of -3 correct? How was the baseline calculated then? I thought that df/F would be limited from below by -1.

`GABAergic ZI neurons are active during hunting¹², feeding²¹, sleep²², fear processing²³, anxiety and novelty²⁴." I think anxiety should be followed by reference 24, and novelty should be followed by something like "exploration" or "investigation" and reference 25. Also relevant might be the Chou et al. Nature Comm 2018 paper on ZIr activity during extinction of defensive behavior.

"Techniplast" should be "Tecniplast"

"40 μ M coronal brain sections" should be "40 μ m ...".

REVIEWER COMMENTS

We greatly appreciate the comments on our manuscript. We performed new experiments and data analysis to address them. The main changes are summarized here.

- New hole-board experiments to link fiber photometry and optogenetic data.
 - Added mZI-CnF stimulation of GABAergic projections.
 - Added CnF VGLUT2 inhibition.
- Electrophysiology experiments.
 - Added in vivo recordings of units in the CnF region following mZI photostimulation to provide evidence for functional inhibition of CnF units.
- Reanalysis of fiber photometry to examine additional exploratory behaviors.
- Ethogram data reanalyzed to add new behaviors and further quantify data
- More data added for mZI-CnF anatomical tracing.
- Added text to interpret and introduce our new findings.
 - In particular we have now defined and interpreted effects on exploratory behaviors separately from locomotion.

Reviewer #1 (Remarks to the Author):

In this manuscript, Sharma et al. examine the role of the inhibitory medial zona incerta (mZI) pathway to the cuneiform nucleus (CnF) in exploratory behavior. Using a combination of optogenetic manipulation, viral tracing experiments, and calcium fiber photometry in freely behaving transgenic mice, they conclude that GABAergic mZI neurons initiate exploratory behavior by inhibiting glutamatergic CnF neurons.

This is a novel pathway and a nice combination of genetic techniques, but the current version of the paper is missing some detail and information about the experiments, lacks a clear definition of foraging behaviors, and does not sufficiently describe neural mechanisms. More importantly, discrepancies in the results disagree with the initial interpretation of data and in contrast suggest that the CnF-mZI pathway would be unlikely involved in foraging and exploration.

Response: Thank you for the comments. We have conducted new behavioral tests and run new sets of experiments. By using a hole-board test, we have provided additional evidence that activation of the mZI to CnF pathway promotes exploratory behavior. Specifically, we observed an increase in the number of head-dips during activation of GABAergic terminals (Figure 6h-i). These findings were supported by parallel experiments in which we inhibited the CnF vGlut2 population where we see more head-dips (Figure 2h-i). We provide more interpretation of mZI fiber photometry data (Figure 4) and emphasize the inverse relationship to locomotor activity, while providing more data supporting activation during exploratory behaviors (rearing & head-dip). We have revised the text and added more detail and analysis to discuss evidence for a role of mZI neurons in exploratory behavior. This has provided a link that was missing between the photometry and optogenetic data. Collectively, these lines of evidence strengthen our hypothesis that mZI-CnF circuit is involved in the promotion of exploratory behavior.

Major concerns:

1. The paper in its current format looks like a short communication without clearly identified sections (introduction, results, and discussion). Given the expertise of the lab in the ZI (Koblinger et al.), it would be important to introduce more background regarding the nature of ZI neurons and their projections in the introduction and discussion. Also, given the importance of behavioral experiments and the small number of figures in the current manuscript, extended data 1, 2, and 3 could be included as figures in the paper.

Response: The reviewer is correct that the original paper was in the form of a brief communication that emphasized brevity and narrative, and did not have the structure of a typical submission. This was

because the original submission was directly transferred, as the Nature Neuroscience editors suggested a transfer to Nature Communications. This process moved the paper through Nature Communications peer review in its original format. In the revised manuscript, we now have the opportunity to expand significantly on the background, methods, and discussion, and furthermore have added new data to address the reviewers' critiques.

2. Behavioral analysis in extended data 1 and 3 is insufficient in the current presentation. Given that statistical analyses of percentages between stimulated versus control animals (S1e versus S1f) will unlikely be significant, the authors should statistically analyze whether photoinhibition of glutamatergic CNF neurons and photostimulation of GABAergic miZI neurons induce changes in the proportion of behaviors in comparison to the prestimulation period for each mouse. The absence of color (white strips) should also be explained. The definition of "in place activity" is too broad and misleading as it represents a combination of different behaviors including grooming, sniffing, stretching, and rearing, of which some are already identified as a class (at least for grooming and rearing). A similar analysis should also be included to study foraging locomotion and locomotion in the hole-board test in Fig. 3.

Response: We have revised the event classifications, where immobility, grooming, rearing, and locomotion remain as categories, and a new category has been introduced to include movements such as sniffing, stretching, turning, turning around, and circling. Similar to other reports in the literature, this class has been labeled as "surveying" and the white space has been incorporated into it. In response to the reviewers' suggestions, we have analyzed the amount of time spent performing each activity and have included these graphs in Figure 1 and Figure 5. Additionally, we have expanded the behaviors presented for photometry to include rearing, immobility, and head-dipping (Figure 4).

3. Figure 1 describes behavior during the open field, elevated plus maze, and light-dark tests upon photoinhibition of glutamatergic neurons of the CnF. The absence of change in locomotor speed upon photoinhibition of glutamatergic neurons of the cuneiform is really surprising and should be carefully explained and discussed in regard to previous genetic studies reporting a decrease in step cycle duration and locomotor speed upon inhibition of glutamatergic neurons of the cuneiform (Caggiano et al., 2018; Josset et al., 2018). The increased number of locomotor bouts in absence of changes in speed might suggest that animals were doing more pauses and more short, faster locomotor bouts. It will therefore be important to calculate the locomotor speed for each locomotor bout to exclude this possibility. Furthermore, the absence of significant differences in bouts of locomotion (Fig. 1f), rearing (Fig. 1i), and time in light chamber (1n) appear really unexpected in regard to the SD.

Please double-check the statistics!

Response: We have clarified the issue regarding the lack of change in speed observed in our study during photoinhibition of CnF vGLUT2 neurons. Firstly, the animals predominantly use slow gaits (walking, trotting) during spontaneous exploration of the open field, and therefore, speeds are restricted to a range that supports these slower gaits. Secondly, Caggiano et al. have reported that higher speeds of 27 cm/s were reduced to 20 cm/s upon inhibition of CnF vGLUT2 without affecting the ability to perform slower gaits. In their study, the high speeds were obtained using airpuff stimulation in a runway. While we can only speculate, it is possible that this approach activated a high number of CnF speed-related vGlut2 neurons, and when the population was inhibited, a reduction in speed was observed. Both control eYFP and eNPHR3.0 mice in our experiments during spontaneous exploration of the open field moved at a mean speed of 9-10 cm/s in baseline before the onset of photoinhibition. Photoinhibition (and photostimulation) is now reported as changes in the proportion of time spent on each behavior. The increase in the number of locomotion bouts is likely due to the reduction in the duration of immobility as seen in Figure 1. We now discuss the reviewers' comment regarding speed in the discussion.

4. Retrograde tracing experiments in Fig. 2 illustrate low and intermediate magnification images of back-labeled GABAergic miZI neurons projecting into the CnF. The quality of images using multiphoton microscopy is great, but it would be useful to quantify the number or density of back-labeled neurons and report the nature of their projections. Are they bilateral or unilateral? The schematic in the extended data S4 gives the impression that the miZI would project contralaterally to the CnF. Is that correct? Furthermore, the retrograde serotype AAVr-ChR2 construct could

have been used in functional studies to assess the role of miZI neurons projecting to the CnF and confirm whether their direct activation modulates locomotor behavior. This photoactivation experiment of miZI neurons would allow the authors to refine the results generated in Figure 4, using direct photoactivation of GABAergic miZI terminals to the CnF, which is not really specific as it labeled neighboring structures such as the PAG and the PPTg (see also my comments in the next section).

Response: We used a retrograde tracing experiment to determine the presence of GABAergic projection neurons in the mZI projecting to the CnF (Figure 3a; n = 2 mice). The qualitative results confirmed the presence of inhibitory mZI GABAergic neurons projecting to the CnF. To quantify the nature of this GABAergic projection in detail and confirm the presence of synaptic puncta in the CnF, we used a technique that selectively labeled presynaptic terminals (Figure 3b; n = 7 mice) and combined it with RNAscope (Figure 3d-i; n = 5 mice), to confirm the presence of GABAergic projections onto vGLUT2 neurons in the CnF. We have analyzed these data further and presented the results in a revised Figure 3. In the manuscript, we have also emphasized the unilateral nature of these projections from mZI GABAergic neurons. The photoactivation experiment of mZI neurons projecting to the CnF using AAVrg-ChR2 is feasible, and activation of mZI neurons is possible. One issue we encountered in adopting this approach is that collateralization of mZI neurons is likely, and for these reasons, we felt that activating terminals directly was preferable. We now discuss the possibilities of activating neighboring nuclei. We have updated our outlining approaches, using Allen data exclusively, and have updated the schematics throughout.

5. The combination of anterograde tracing with RNAscope in Fig. 2 to identify the phenotype of CnF neurons is an elegant approach, but the extent of labeling raises some concerns as it is not circumscribed to the CnF; it also projects to the dorsolateral PAG and the lower part of the SCP, suggesting that it might also contact parts of the PPTg. Given the role of these last nuclei in locomotor behavior and exploratory behavior for the PPTg (Caggiano et al., 2018), it will be important to quantify and illustrate the extent of these terminals at low magnification at the brain level and at high magnification at the nuclei level.

Response: The mZI GABAergic neurons have been reported to innervate the PAG¹, SC and PPN², and our data also confirm these observations. We have now added the images showing presence of SYP-mRuby puncta in these neighboring regions (Supplementary Figure 1). We acknowledge in the discussion that there may be activation of neighbouring areas with photostimulation. We added additional data using RNAscope to augment our findings on the mZI-CnF circuit (Figure 3).

There are several lines of evidence suggesting that the CnF is contributing to the observed behavior. Firstly, we included mice where the optic fiber was over the CnF, limiting activation of neighboring areas. Secondly, we used anti-ChAT antibody to label cholinergic neurons in PPN to ascertain the boundaries of PPN and CnF; our injections were found to be on target centered on the CnF, sparing the ChAT labeled PPN region. Both of these observations would suggest a bias towards CnF neurons being inhibited. Thirdly, the functional effects of mZI to other nuclei would not be expected to be similar to our reported effects. An earlier study (Hormigo et al., 2020) showed that ZI activation of GABAergic neuronal projections in the PPN promotes avoidance behavior, yet we found that exploratory activity was promoted. Another study has reported that activation of PPN vGLUT2 neurons produced exploratory behavior, and inhibition would be expected to decrease exploratory behavior. Fourthly, our new electrophysiology data show that a subpopulation of CnF neurons decrease spiking at monosynaptic latencies following mZI photoactivation. The electrophysiological data qualitatively match the anatomical data which also show evidence for a subpopulation of CnF neurons being targeted. Finally, although dPAG is a possibility, it lies rostral to our optrode, limiting the effects. Taken together, these findings suggest that the mZI-CnF circuit being activated by terminal stimulation is a reasonable conclusion.

6. The density of synaptic puncta of VGAT miZI neurons onto VGLUT2 CNF neurons should be quantified. The authors should discuss their results regarding the presence of 40% of non VGLUT2 (presumably GABAergic) CNF

neurons that receive GABAergic mZI inputs. Could it be a disinhibitory miZI-CnF pathway? If so, that's a novelty and it should be mentioned. The legend of Fig. 2e-f should be corrected. It should be clarified whether images were collected using multiphoton or confocal microscopy in legends and methods. The use of machine learning is interesting, but it is not clear how it was used. Was there any validation of the predictions?

Response: We have now clarified that we did not obtain multiphoton data - only confocal data. We have quantified the density of synaptic puncta of VGAT mZI neurons onto weak (4-9 puncta), medium (10-24 puncta), and strong VGLUT2 CnF (25 or more puncta) neurons. The non-vGLUT2 population with SYP-mRuby puncta is likely CnF GABAergic neurons. We have discussed the possibility of a disinhibitory mZI-CnF pathway, but future studies are required to confirm this with appropriately designed experiments. We have updated Figure 3 (previous Figure 2) with additional analysis. The machine learning approach was via ilastik segmentation, which has been published⁴. We have validated the published approach independently of this paper, and we have now included the appropriate citations in the current manuscript.

7. The definition of foraging locomotion in Fig. 3 is problematic. How can animals forage and walk at the same time? One can expect animals to move around their food or an object while foraging or exploring, but can we really call that locomotion? Using the behavioral classification (extended data 1 and 3), the authors should be able to more specifically distinguish foraging behaviors from locomotion.

Response: We have removed the term 'foraging' from the manuscript as it is not reflect our behavioral tests. We define locomotion as requiring the animal to move from point A to B in a given space with a defined speed and displacement in a unit time. Using this definition of locomotion, we have separated walking bouts in the hole-board arena into two distinct groups and also controlled for locomotor velocities: (1) moving around the corners of the arena and (2) bouts of locomotion performed moving between one hole to the next while engaging in head-dips. We have termed these thigmotactic locomotion (Figure 4g) and focused exploration (Figure 4h), respectively. We have adopted terminology used in previous publications⁵. Our data shows a higher activation of mZI neurons when mice are engaged in focused exploration versus thigmotactic locomotion when controlled for velocity (Figure 4k). The findings are now further supported by our new hole-board data where activation of mZI to cuneiform GABA terminals causes an increase in head-dip behavior (Figure 6h-i), as predicted by our photometry data.

8. There are some discrepancies in Fig. 3 and S2. How can calcium activity of GABAergic miZI neurons be increased immediately prior to initiation of foraging locomotion in Fig. 3, but also be increased about 1 second after the initiation of head-dip in Fig. S2? In all cases, profiles of calcium transients in Fig. 3 should be analyzed for all behavioral conditions during the hole-board as well as in the open field. Calcium transients should also be analyzed as a function of the locomotor speed.

We acknowledge the reviewer's concern around calcium transients and locomotion. Our further analysis illustrates that mZI activity is inversely related to locomotion onset and speed (Figure 4g). When we anchor it to the start of the head-dip, it starts to increase at the onset of head-dip and this sustained elevation peaks 1.2 seconds later (Figure 4f). We also reanalyzed the data now anchored to the end of the head-dip (see below). As predicted, this precedes slightly the 0 point and drops during locomotor onset.

Given earlier studies showing that MLR evoked locomotion is under inhibitory GABAergic control, the inverse relationship between mZI GABA activity, locomotion onset, and speed during focused exploration is what would be predicted. It is also noteworthy that mZI GABAergic neuronal activity is greater for certain types of locomotion (Figure 4j), suggesting a strong context-specific modulation. Additionally, ZI GABAergic neurons have been shown to have heterogeneity in terms of their involvement in context-specific movements and locomotor behaviors. To exclude the possibility of activating a heterogeneous population of mZI GABAergic neurons with diverse brainwide projections, we chose to activate mZI-CnF terminals. Indeed, our new experiments using mZI-CnF GABAergic terminal stimulation increased exploratory head-dipping behavior (Figure 6h-i) and an increase in exploratory behaviors was

found upon inhibition of CnF vGLUT2 neurons (Figure 2h-i). Collectively, these observations highlight the specificity of the mZI-CnF circuit in the promotion of exploratory behavior.

We have provided more profiles for hole-board analysis, including rearing (Figure 4e) and immobility (Figure 4d), to compare with head-dip behavior. We found an increase in calcium transients during rearing (Figure 4e) but not for immobility (Figure 4d). This is interesting because mice stay in place during rearing and especially hole-board activity. So it appears specific to the task itself.

Concerning potential technical biases, would it be possible to confirm that calcium activity is unrelated to the animal's head movement while head-dipping in the hole?

Response. We examined the traces where mice engaged in a head-dip. The change in the 470nm signal showed variability during the head-dip event (see example trace in Figure 4 and example below). We would have expected less variability if it were due to mechanical effects. Also, the fluorescence rose during consecutive head-dips, yet the amplitude of the event was the same (Figure 4). This would be unexpected if motion artifacts in the isobestic 405 nm reference signal were removed.

Still, there remains a possibility that mechanical movement of the head caused substantial changes in the signal. To test this, we examined traces from mice in the 405 nm isobestic signal during head-dips. This reflects movement artifacts alone. When this was done, there was no appreciable difference in the reference 405 nm signal (in red) during head-dips (see trace below) compared to the 470 nm signal. Overall, we are confident that the signals reflect calcium transients and not movement artifacts

Furthermore, although the hole-board test is used as an exploratory test, it has also been considered as an escape behavior in some studies (Brown and Nemes, 2008)! Perhaps this should be mentioned and discussed.

We recognize that there is some debate in the literature in terms of what the hole-board is measuring. The idea that head-dipping is an escape behavior is an interesting possibility and should be considered. However, it is not universally accepted, and the research was completed in rats. When we look at work where the cuneiform is stimulated, which is associated with escape-like behaviors, we see a decrease in hole-board head dips³. We are aware of downstream functional connections from the cuneiform to nuclei associated with emotional behaviors such as the amygdala⁶. It is indeed possible that increasing GABA drive to the cuneiform would reduce amygdala activity, and be anxiolytic, as the cuneiform has ascending projections to the amygdala. This is an exciting new direction for MLR research as we start to integrate locomotor initiation with the emotional needs of the animal. We now discuss this.

9. In absence of changes in locomotor speed (Fig. 4e) and bouts (4f), the increase in distance (4g) and time (4h) traveled in the center of the open field as well as the number of rearings (4i) upon PS of GABAergic miZI terminals in the CNF is a bit surprising and unexpected. Can the authors explain? Moreover, the large SD in Fig. 4h suggests that the time spent in the center of the open field is unlikely significant. Please double-check the statistics.

Response: We have also addressed this issue in comment 3 above. In summary, mice in the open field displayed a range of speeds closer to slower gaits, similar to walking and trotting in spontaneous exploration of an open field. As far as the likelihood of an increase in locomotion distance without a change in speed is concerned, it can occur in the open field either by increasing the number of locomotion bouts or by increasing the duration of locomotion bouts. Here, an increase in the duration of locomotion and corresponding increase in distance traveled have been reported without a change in speed and number of locomotion bouts. We have double checked the statistics for 4h (Figure 5 j) and there is a main effect of time using two-way repeated-measures ANOVA; Time: $F_{(1,80,32,41)} = 6.95, p = 0.0040$ and when compared baseline Pre vs light ON conditions using *post hoc* vGAT^{ChR2} Pre vs light ON; $p = 0.012$ was observed in vGAT^{ChR2} mice in laser ON condition whereas no effect was seen in vGAT^{eYFP}

10. I have some difficulties in reconciling the increased calcium activity (Fig. 3g) of GABAergic miZI neurons during the “pause” prior to initiation of “foraging locomotion” (Fig. 3f) and the absence of change in locomotor speed (Fig. 1e and 4e) in response to optogenetic inhibition of glutamatergic CNF neurons or optogenetic activation of GABAergic miZI terminals in the CNF. This is inconsistent and suggests otherwise that the CnF-miZI pathway would be unlikely involved in foraging or exploration. Check Caggiano (2018) for DREADD inhibition - animals still move.

Response: The activity of the mZI is inversely related to speed in a context-specific manner and we now make this point clear in the manuscript. mZI activity levels increase during head-dip and rearing events. We have now provided evidence that photoinhibition of CnF vGLUT2 and photostimulation of mZI vGAT terminals in the CnF increase head-dip behaviors in the hole-board (Figure 2 and Figure 6), which adds additional support to the conclusion that GABAergic modulation of the CnF via the mZI circuit contributes to exploratory behavior. These findings support each other.

We understand that the lack of reduction in speed following photostimulation of mZI fibers or photoinhibition of vGLUT2 is counterintuitive. We have also discussed the change in speed during open field experiments in comments 3 and 9 above. The findings of Caggiano et al.,³ used DREADD inhibition of vGLUT2 CnF. Airpuffs were used to elicit higher speeds of locomotion in the walkway. They were able to reduce speed from 27 cm/s to 20 cm/s without affecting slower speed gaits like walking and trotting. This is consistent with our findings, where we focused on spontaneous locomotion. Mice rarely use gallop

or bound gaits during spontaneous exploration of the open field and generally use walking and trotting gaits in the open field.

Minor comments:

Legend of Figure 2:

2d: Rephrase: presence of synaptic puncta (in magenta) of mZI inhibitory fibers “in the CnF” innervating CnF glutamatergic vGLUT2 neurons (in green) showing presence of vGLUT2 mRNA.

Response: A correction has been made in the revised manuscript.

2e: Cells in the CnF cannot express but colocalize with SYP-mRUBY synaptic puncta of GABAergic mZI terminals. The label and legend in Fig. 2e are not clear: What does the author mean by vGLUT2 neuronal projections in VGAT-CRE mice?

Response: A correction has been made in the revised manuscript.

Statistics: I assume that the stats represent the mean +/- SD. Please mention this in the figures or methods.

Response: A correction has been made in the revised manuscript in the methods section to indicate that stats represents mean +/- SEM.

Page 2: Recent studies have shown (only one study is referenced).

Response: A correction has been made in the revised manuscript.

Page 3: VGAT2ChR2 should be VAGTChR2

Response: We have checked for the consistency in using the term “vGATChR2”.

Reviewer #2 (Remarks to the Author):

This is an interesting study. In this study, they found a new ZI-CNF circuitry for claiming the role in exploratory behaviors. However, it seems that their experimental results cannot clearly support their claims in the current version of the manuscript. I think that the authors need substantial revisions with additional experiments to clarify their claims.

First, the definition of exploratory behaviors they suggested is not clear. They described “These results provide evidence for context-dependent dynamic regulation of CnF vGLUT2 neurons, with the mZI-CnF circuit providing an important contribution in foraging and exploratory behavior.” (page 1). For examining the causal role of ZI-CNF circuitry for exploratory behaviors, they did open-field test, elevated plus maze and light/dark test with optogenetic excitation/inhibition experiments. In the previous many papers, open-field (calculating the duration in center zone), elevated plus maze and light/dark test have been used for claiming anxiety. Even though the complete, conceptual separation of exploration and anxiety is quite difficult in experimental conditions, they need to sophisticatedly perform more additional experiments for explorative behaviors like hole-board test (the authors already did for fiber-photometry in Fig.3) or other types of exploratory behaviors with optogenetic excitation/inhibitions.

Response: We have now revised the definition of exploratory behavior and omitted the word 'foraging' from our interpretations as it was also suggested by another reviewer. We have performed additional optogenetic experiments to confirm exploratory behavior using a hole-board, which complements and extends the conclusions drawn from the open field data in our original submission. The results of these experiments are now included in the manuscript (Figure 2 and Figure 6).

The reviewer raises an excellent point that the data could be interpreted as the cuneiform nucleus output reducing anxiety-linked behavior. However, since an increase in exploratory locomotion is observed in another arena, it suggests that a reduction in anxiety is not the primary behavioral response. We have

now addressed the reviewer's points regarding anxiolytic effects in the discussion. We feel that this shines a light on the intricacy of cuneiform output. It is not just a regulator of locomotion and may be embedded in the control of affective behavior (fear, anxiety, etc.).

Second, they did hole board test only for fiber-photometry. There are logical gaps in the experimental design and the interpretation of the role of ZI-CnF circuitry in anxiety-like behaviors (Fig. 1 & 4) or hole board explorative behaviors (Fig. 3). The authors need to clarify the mechanistic link between the results from Fig1, Fig4 and from Fig.3.

Response: We have now performed holeboard experiments for both photoinhibition of CnF vGLUT2 neurons and photostimulation of GABAergic mZI-CnF terminals. We have provided more text in the results section to connect the approaches and tests used.

Third, in the paper of Caggiano et al., (2018, Nature, <https://www.nature.com/articles/nature25448>), they show the chemogenetic inhibition of CNF_vglut2 neurons does not affect the exploration with hole board tests. How do the authors interpret these results considering the author results?

Response: We have reviewed the experiments performed in the paper by Caggiano et al., 2018. They used photostimulation to activate CnF and PPN vGLUT2 neurons in the hole-board but used DREADD approaches to inhibit the population. They reported that 10s long epochs of VGLUT2 photostimulation strongly suppress any further locomotor activity in the hole-board. Such changes in behavioral state are hallmarks of a defensive reaction often seen after the presentation of threat stimuli such as electric shock, predator, etc. Indeed, such a state, termed defensive arousal, was reported after brief photostimulation of VMH neurons evoking escape behavior⁷. On the other hand, competing survival behaviors such as hunger can promote exploration and foraging behaviors via suppression of neurons and circuits involved in defensive behavior, even in the presence of predation threat in the environment. Since CnF vGLUT2 neurons promote defensive behavior, therefore inhibition of these neurons promoting increased hole-board exploratory behavior in our work can be explained in light of competing circuits that are both essential for survival. The difference between the results could be a result of the different methodology used. DREADD approaches in Caggiano may have resulted in desensitization over time, for example. By the same token, synchronous inhibition of vGlut2 neurons by photoinhibition may also contribute to differences observed.

Reviewer #3 (Remarks to the Author):

In the manuscript "Inhibitory medial zona incerta pathway drives exploratory behavior by inhibiting glutamatergic cuneiform neurons", the authors show that inhibition of glutamatergic neurons in the cuneiform nucleus (CnF) leads mice to spend more time in the center of an open field box and in the open arms of an elevated plus maze. They found that GABAergic neurons from the mZI project to the CnF and are active during exploratory behavior, in particular head-dipping during hole exploration. Activation of the mZI to CnF projection results in similar exploratory behavior as inhibition of the glutamatergic CnF neurons. The manuscript concludes that this inhibitory projection promotes exploratory behavior. This adds an interesting new finding to recent work on the roles of the CnF and the ZI in motivated behaviors.

This conclusion is supported by the data and previous literature. In particular, because locomotion by itself does not seem strongly affected by the optogenetic manipulations, although the manuscript is a little unclear about this point. I believe that the experiments by themselves also allow the conclusion that the primary action is a reduction in anxiety, indirectly leading to an increase of exploratory behavior. Previous literature on the CnF and mZI, however, support the conclusion reached in the manuscript that the projection drives an increase in exploration, independent of anxiolytic effects.

Response: Thank you for the feedback. We have taken the comments into consideration and have revised our manuscript accordingly. We have strengthened our conclusions regarding the circuit being

involved in promoting exploratory locomotion by adding holeboard experiments. We agree that a reduction in CnF output could be anxiolytic, but believe that our new data support the idea that exploratory behavior is promoted. The CnF does have connections to the amygdala and this is a topic of interest in terms of the CnF being a driver of emotional behavioral responses along with its traditional motor control role. We have added this topic to our discussion.

One surprising omission in the presented experiments is the combination of the hole-board test with activation and/or inactivation of the mZI to CnF projection. This would show more clearly the exploratory nature of the behavior. The fiber photometry for the hole-board test has been done for the mZI neurons, and not for the CnF neurons or the mZI to CnF projection (if the latter is possible). The tests presented for the mZI to CnF are a little less specific for exploratory behavior. I am reluctant to ask to this whole new set of surgeries and experiment, but it would make the evidence for the conclusion much stronger.

Response: We have now performed additional experiments using a holeboard in combination with optogenetic modulation. Both terminal stimulation of mZI GABAergic fibers at the level of the cuneiform and halorhodopsin approaches to silence vGLUT2 neurons in the CnF were used. The results of these experiments, which are now included in the manuscript (Figure 2 and Figure 6), show that photostimulation of the mZI terminals at the CnF increase rearing and head-dips. We also show that VGLUT2 inhibition had broadly similar results with a significant increase in head-dips. These new data add more evidence to support our conclusion that inhibitory modulation of the CnF contributes to mice exhibiting exploratory behaviors.

Besides this point, I only have a number of small comments (roughly in order in which they appear in the manuscript).

In Fig 1b-c, the dorsal atlas outline does not fit the slice at all, and with the image quality of the review manuscript, it is difficult to judge whether the real slice is close to the -4.84 mm atlas slice that the projected CnF outline is close to the real outline of the CnF. Is it perhaps possible to adjust the image settings or upload a better quality picture to allow the reviewer, and the reader, to better judge the accuracy of the fiber placement?

Response: In the revised manuscript we have updated Figure 1b with a matching atlas slice (Allen Brain Coronal Reference Atlas). The revised figures clearly illustrate the position of fiber above CnF vGLUT2 neurons expressing eNPHR3.0 (in green) and the ventral PPN marked with the presence of cholinergic neurons (in magenta) shows lack of eNPHR3.0 labeled neurons.

Fig 1b-c. Please add which atlas outline is used.

Response: In the revised manuscript we have updated the information on the coronal reference atlas used in the figures (Allen Brain Coronal Reference Atlas).

Fig 1f. The corresponding caption and text suggests significant change in number of locomotion bouts, but no star is shown. Is this right?

Response: The increase was not significant so to remove confusion we have now revised the sentence to "Total number of locomotion bouts do not show significant increase in vGLUT2^{eNPHR3.0} mice during laser ON epoch."

Fig 1i. The corresponding caption suggests significant change in number of rearings, but no star is shown. Is this right?

Response: We have now presented a detailed analysis comparing proportions of change in events before laser and during laser in each mouse as this was suggested by another reviewer.

Fig. 1. Is the number of animals written somewhere? I cannot find it. If not, please add number of animals to captions, corresponding text or elsewhere.

Response: The number of animals used is added to the figure captions in the revised manuscript.

Fig. 1n. No star is shown. Is this correct?

Response: it is correct.

“Our results that show the inhibition of CnF vGLUT2 neurons increasing exploratory behavior proposes the CnF as a critical part of the defensive behavior circuit” This seems too strong a conclusion at this point, because inhibition of these neurons has shown effects that could also still be interpreted as a drop in anxiety. I would suggest phrasing it less strongly.

Response: We have revised the text as suggested and phrased it less strongly. We have acknowledged and discussed the possibility that the CnF may play a role in reducing anxiety, as well as priming motor circuits for escape-like locomotion, while promoting exploratory behavior in low-anxiety conditions. We have included this in the discussion and acknowledged that it may be part of the cuneiform's role in contributing to setting emotional states, in addition to its traditional role in motor control.

Fig. 2a. Please mention how the ZI outline was determined and drawn.

Response: The image is matched to a comparable slice from allen brain atlas image. The adapted atlas outline is used to draw the ZI outline.

“Consistent with other reports”. Please add references to these reports.

Response: References have been added as suggested.

Fig. 2c. What is the relationship between the rectangular region outlined in the left panel and the square images on the right?

Response: it is now described (revised Figure 3c)

Fig. 2d. What do the arrow heads mean?

Response: a description of the arrowheads is added (revised figure 3d).

“Our results show that CnF cells are predominantly glutamatergic (Figure 2e: Mean = 65.9 ± 5.21 ; $p = 0.0403$),” What does this p-value mean? What is tested? Is this a useful test? What are the dots, slices or mice?

Response: We have now highlighted in the revised manuscript that we tested the distribution of vGLUT2 neurons and non-vGLUT2 neurons in the CnF using unpaired t-test ($n = 5$ mice).

“nearly half of the vGLUT+ cells were associated with SYP-mRUBY synaptic puncta” Indeed, and given the larger number of VGLUT+ cells, I am very willing to believe that they are recipient of the input from the mZI, but the image Fig 2d does not really show very well that all these associated SYP-mRUBY puncta indeed contact the VGLUT+ positive cells. The word “confirm” in the line “These results confirm the presence of a mZI-CnF circuit providing inhibitory input to the excitatory vGLUT2+ cells” is a bit strong, something like “strongly suggest” is more accurate, I feel. Patching vglut2-cells and (optogenetic) activation of mZI cells/projections would be the way to really confirm this.

Response: This is a good point. To address the issue we performed in vivo electrophysiological experiments where we photostimulated the mZI GABAergic population and recorded from the CnF region. We found that a proportion of neurons is inhibited at monosynaptic latencies (Supplementary Figure 3).

“(Figure 2e: Mean = 49.15 ± 51.6 ; $p = 0.8774$).” Is this a useful test?

Response: We have now updated this figure with a new figure (Figure 3) with revised results.

Fig. 3 e . Is a df/F of -3 correct? How was the baseline calculated then? I thought that df/F would be limited from below by -1.

Response: Z-score was used for our normalization calculation for our fiber photometry data. There was an error with the y-axis titles and this is corrected. Hence, the normalization calculation we used was

(signal - mean) /stdev. We went and looked over our FP data and did not see any signals reach a value of -3.

‘GABAergic ZI neurons are active during hunting¹², feeding²¹, sleep²², fear processing²³, anxiety and novelty²⁴.’ I think anxiety should be followed by reference 24, and novelty should be followed by something like “exploration” or “investigation” and reference 25. Also relevant might be the Chou et al. Nature Comm 2018 paper on ZIr activity during extinction of defensive behavior.

Response: A correction has been made in the revised version.

“Techniplast” should be “Tecniplast”

Response: A correction has been made in the revised version.

“40 μM coronal brain sections” should be “40 μm ...”.

Response: A correction has been made in the revised version.

REFERENCES

1. Chou, X.-L. *et al.* Inhibitory gain modulation of defense behaviors by zona incerta. *Nat. Commun.* **9**, 1151 (2018).
2. Hormigo, S., Zhou, J. & Castro-Alamancos, M. A. Zona Incerta GABAergic Output Controls a Signaled Locomotor Action in the Midbrain Tegmentum. *eNeuro* **7**, (2020).
3. Caggiano, V. *et al.* Midbrain circuits that set locomotor speed and gait selection. *Nature* **553**, 455–460 (2018).
4. Berg, S. *et al.* ilastik: interactive machine learning for (bio)image analysis. *Nat. Methods* **16**, 1226–1232 (2019).
5. Casarrubea, M., Sorbera, F., Santangelo, A. & Crescimanno, G. Microstructure of rat behavioral response to anxiety in hole-board. *Neurosci. Lett.* **481**, 82–87 (2010).
6. Lam, W. & Verberne, A. J. Cuneiform nucleus stimulation-induced sympathoexcitation: role of adrenoceptors, excitatory amino acid and serotonin receptors in rat spinal cord. *Brain Res.* **757**, 191–201 (1997).
7. Kunwar, P. S. *et al.* Ventromedial hypothalamic neurons control a defensive emotion state. *Elife* **4**, (2015).

REVIEWER COMMENTS

Reviewer #1 (Remarks to the Author):

The paper by Sharma et al. investigates the functional role of a new pathway connecting the Zona Incerta (ZI) to the cuneiform nucleus (CnF) in exploration. The authors have included some new anatomical and functional studies and reorganized and improved the current version of the manuscript in response to previous comments, but there are still some problems.

The authors show that the activity of GABAergic neurons of the ZI is upregulated during rearing and during the hole-board test. They also show that photostimulation of GABAergic ZI neurons inhibits the neural activity of a small population of the CnF. They also characterize the anatomical pathway connecting the ZI to the CnF. However, there is no information about other midbrain/brainstem targets of the ZI that are mentioned in the discussion (for their role in modulating exploratory behaviors) but never in the introduction. Behavioral data supporting exploration (hole-board test and rearing) are unfortunately overshadowed by other tests (light-dark, elevated plus maze) that did not explicitly test it, and instead suggest an anxiolytic effect on behavior as mentioned earlier by the 3rd reviewer.

From the current results, we cannot exclude that the mZI-CnF pathway contributes to some of the behavioral changes in exploration, but the lack of changes in the ethograms suggest instead that this pathway plays at best a minor role.

Given the projection pattern and modulatory role of GABAergic neurons of the ZI onto the PAG and PPN, it could be important to investigate these other targets in parallel to the CnF. Alternatively or in combination, using a retrograde photostimulation approach of CnF-projecting ZI neurons could also strengthen the authors' conclusion regarding the role of this ZI-CnF pathway in exploration.

Major comments

1. Glutamatergic neurons of the CnF do not only induce high-speed gait. Several studies have shown that photostimulation of glutamatergic neurons of the CnF induces a wide range of locomotor gait and speed, not only an escape or flight reaction. Chemogenetic inhibition as well as photoinhibition of glutamatergic neurons of the CnF can modulate control of slower speed. Therefore, using the rationale that the CnF only induces high-speed gaits is a misleading simplification.

2. Behavior: An increased proportion of locomotor bouts and an increased time spent in the center of the open field, as shown in Figs. 1 and 5, do not necessarily reflect an increase in exploratory behavior. Similarly, an increased activity in open arms of the elevated plus maze test or in the light chamber in the light-dark test (Fig. 2 and 6) might simply reflect a decreased anxiety in normal behaviors, thus pushing animals to expose themselves fearlessly in open areas. These results thus do not reflect exploration, but could be considered as a prerequisite to exploration (i.e., a pre-exploratory condition). Although it has been previously reported as an escape test (as mentioned by the authors), only the hole-board test, and to some extent the rearing test, should be considered as real exploratory tasks.

3. Despite some statistically significant changes, the ethograms do not exhibit a clear pattern—either for the immobility in Fig. 1e-f or for the rearing in Fig. 5e-f in optogenetically manipulated mice. Overall, there is a trend with an increase in surveying, rearing, and locomotion upon optogenetic manipulation in comparison to controls in Fig. 1g or 5g. However, the effects appear to be subtle with only a decrease in immobility upon inhibition of the CnF and an increase in rearing upon stimulation of their GABAergic inputs. These minor changes in ethograms suggest, therefore, that the ZI-CnF pathway plays a minor role in exploration.

Would it be worth analyzing several trials per mouse (rather than a single trial per mouse) and a higher number of mice to reveal stronger changes? Using naïve mice rather than accustomed ones could also increase the probability of exploratory behaviors. As shown in previous studies, bilateral optogenetic manipulation might also be more efficient than unilateral stimulation in generating behavioral changes in the ethograms.

4. I appreciate the quantification of SYP-mRuby+ VGAT+ ZI boutons onto VGluT2+ CnF neurons. RNA scope is known to impair cell morphology, and I am pleased to see that the authors used DAPI to delineate the extent of the cells. Although I have a few reservations about the

quantification of synaptic boutons onto largely deformed cells, I assume the analysis is representative of the observation. Nevertheless, the legend and the results section describing VGluT2 puncta and SYP-mRuby synaptic puncta quantification are confusing. I am not quite sure I understand the rationale for quantifying VGluT2 puncta in VGluT2 positive CnF neurons from the RNAscope study. What does it add to the study?

5. Supplementary Figure 1 raises questions about the connectome of the ZI pathway. It is still not clear whether the ZI project bilaterally or unilaterally. According to previous tracing studies (Caggiano et al., 2018 and Dautan et al., 2021), the PPN receives far more ZI inputs compared with the CnF. And stimulation of GABAergic ZI terminals in the PPN suppresses active avoidance (Hormigo et al., 2020), thus potentially contributing to motor activity and perhaps exploration. It is also well known that the PAG is involved in strong escape and flight reactions, and it receives numerous inputs from the ZI, in addition to project onto the CnF. Taken together, these findings suggest that the PPN and PAG could be potentially more pertinent functional targets than the CnF of GABAergic ZI neurons to exploratory behaviors.

6. I am not sure I understand why the authors cannot use a retrograde approach to stimulate ZI neurons projecting to the CnF. If photostimulation of GABAergic ZI terminals onto the CnF suggests that this pathway could be involved in exploration, there is also a high risk of activating ZI collaterals projecting onto neighboring nuclei (i.e., the PAG and the PPN). In contrast, a retrograde approach to stimulate GABAergic ZI neurons projecting to the CnF would allow the authors to more specifically address the contribution of this pathway to exploration.

Minor comments

Introduction

Page 1, line 7, replace with more recent reviews.

Page 1, 2nd paragraph

Replace the reviews with scientific articles in "The CnF has connectivity with the sympathetic system."

Page 1, 1st paragraph

CnF neurons are also involved in walking gaits, not just high-speed locomotion (Fig. 1J of Caggiano et al., 2018, and Fig. 6C of Josset et al., 2018). CnF is not incompatible with slower locomotion, and this should be clearly stated. Same for the discussion at page 7.

Figure 1

Ethograms in Figure 1e-f. The main finding regarding behavior in Fig 1g is less immobility. When looking at the ethograms, one mouse is mostly immobile. One clear thing is the lack of effect on behavior, which is confirmed in Fig. 1g. This is odd because mice show more locomotor bouts when CnF is inhibited (Fig. 1i). Also, what is shown in those ethograms, a single trial for each mouse? What is the bin size? Please state these details in the legend. Have you performed several trials? If yes, what is the proportion of each behavior?

Page 2, line 2

This study is a non-peer-reviewed article and it does not show any evidence of interconnection between both nuclei. I suggest replacing the reference with a peer-reviewed paper and/or correct the statement about connectivity throughout the text.

Page 3, 1st paragraph

Several statements without statistical analysis should be removed from the paper. "During spontaneous exploration... largely restricted to walking and trotting gaits." "We observed that control... showed a time-dependent reduction in locomotion." In absence of gait and statistical analyses, these are only qualitative and speculative.

Page 3, 3rd paragraph

"with the exception of few studies... head-dip activity in the hole-board." The interpretation of the hole-board test as an exploratory test and its potential assessment as a measure of escape should be discussed in the discussion rather than in the results section.

Page 4, conclusion at the top. Unilateral inhibition of the CnF does not suppress locomotor behavior according to your data. Remove or rephrase.

Supplementary Figure 1

Without quantification, it is difficult to concur with a unilateral bias. Furthermore, it shows a labelling as dense in the PAG and PPN, supporting the need to investigate these pathways.

Page 4. "Our data also shows that mZI GABAergic projections are biased towards a subpopulation of CnF vGLUT2 neurons (Figure 3i)." Maybe rephrase because I do not see this bias. Also, it seems arbitrary to use these three categories of puncta density. Why not simply report the distribution of puncta per cell?

Page 4, line 1,

Again, correct the statement: the CnF is not only correlated with high-speed locomotion.

Page 4, 1st paragraph

"Monosynaptic retrograde... neurons¹⁴." Add Caggiano et al., 2018 as a reference.

Page 4, 2nd paragraph

The verb "to demonstrate" is used several times throughout the draft. It is improper in the current context. Please correct.

"A recent study...vglut2 mRNA puncta" should be moved to the discussion.

Figure 3

The heatmap of reporter mGFP is not necessary in Fig. 3b.

There is a high density of ZI terminals onto the PAG and a smaller patch in the CnF in Fig. 3c. It would be great to show the axonal terminal pattern of ZI onto the PAG, CnF, and PPN in more rostral sections. Quantification will be important.

Figure 4.

Open field locomotion and ZI activation are not related but the current z-score scaling overemphasize the small 0.2 peak at the onset of locomotion. I suggest that the authors match the scaling of panel n with the one of panel j for better clarity of results.

Page 5, 1st paragraph

Supplementary Figure 3 on multi-unit recordings of CnF neurons upon photostimulation of the VGAT+ ZI neurons adds to the comprehension of the results, but it should be integrated into the paper and better presented. It is not clear from the results section how many cells were recorded from 3 mice. The firing frequency of CnF neurons, the total number of recorded multi-units, and the number of depressed cells upon ZI photostimulation should be reported.

In panel 3c, why is there a category for "increase" in the legend but not in the pie chart itself? What are the statistical criteria mentioned in the methodology to define a responding (inhibited) cell? One can clearly see inhibition, but there is no clear evidence of a post-inhibitory rebound activity.

Please correct "demonstrate."

"Collectively these data...".

Page 5, 2nd paragraph

Add Caggiano et al., 2018 as ref to ref 46-47 about head-dipping.

The increased calcium activity in VGAT ZI neurons during rearing and head-dipping is surprising and weird. How can the same neuronal subpopulation generate at once a similar activity for two distinct exploratory behaviors? It is not clear to me.

Some interpretations of the results are speculative and should be transferred to the discussion

Page 5, "Furthermore.... ...during escape behavior.²¹"

Page 6, 1st paragraph, "These results highlight a role of ZI GABA... ... novelty-seeking.⁵²"

Figure 5

In comparison to supp. Figure 1, I am a bit surprised by the low expression of terminals in the CnF and the absence of ChR2-YFP expression in the PAG or PPN from this tracing in Figures 1 and 5. The tip of the optical fiber is far away from the ChR2-YFP expressing ZI terminals in the CnF in Fig. 5. Was it sufficient to activate GABAergic terminals?
Fig. 5e does not show any striking pattern, despite statistical differences in rearing in Fig. 5g.

Although CnF-projecting VGAT+ZI terminals do not affect immobility, they seem to increase motor activity (i.e., rearing). It seems that they also tend to increase locomotor activity. Maybe when assessing more than one trial per mouse these effects could be significant?

The increased trend in travelled distance and time upon stimulation of VGAT+ZI terminals of YFP expressing mice in Fig. 5j-k is a bit puzzling. This raises questions about the efficacy of blue light to induce behavioral changes in wild-type mice in the absence of ChR2. Fig. 6 is more convincing than Fig. 5a-g by showing a clear effect in the elevated plus maze test, the light-dark test, and the hole-board test with a good control group showing very few changes upon stimulation.

Supplementary Figure 1

The outline should be aligned with the brain section image. There is no clear difference at first sight in the axonal terminal patterns onto the CnF, PPN, and PAG. There is still a question about the lateralization of this ZI pathway. Some quantification will be necessary.

Supp. Fig. 4 is not necessary and misleading as the CnF does not generate only escape or flight behavior.

Discussion

Page 7

Add Caggiano et al., 2018 to ref 14.

Correct last sentence in regard to the CnF or PPN modulation of slow locomotion (Caggiano et al., 2018).

Page 8, 1 line, correct the statement

Page 8, 3rd paragraph should be reorganized; studies that investigated PPN and PAG populations receiving ZI inputs should be better presented and in context with the current study (Ferreira-Pinto et al., 2021; Chou et al., 2018). The use of the MLR instead of the PPN regarding the Ferreira-Pinto et al., 2021 study is misleading and could give the impression that they studied distinct CnF-related pathways, while it was actually the PPN.

Page 8: "These observations suggest that CnF inhibition contributes to postural adjustments and the suppression of competing motor pathways (e.g.: walking, rearing, grooming, etc.)." There are no statistical changes to support this conclusion.

Page 9: "...mZI terminals could activate neighboring nuclei..." Do you mean inhibit?

Reviewer #2 (Remarks to the Author):

The authors addressed all issues I raised about adding several experiments. However, I think that the authors need to edit some parts of the main text for clarifying some points as below.

1. In the last paragraph of page 2, I cannot understand why the authors put the sentence 'The behavioral repertoire of mice is sensitive to environmental features' in the middle of this paragraph.
2. In the second paragraph of page 3, the authors need to add some words after 'To confirm' to clarify the purpose of this experiment
3. In page 3, the third paragraph is not clear

4. In the first paragraph of page 6, I think that the authors cannot say that mZI GABAergic neurons modulate fear generalization, anxiety, and novelty-seeking in this paragraph. They just show the neural correlates of mZI inhibitory vGAT neurons

Reviewer #3 (Remarks to the Author):

The authors have done extra experiments, which all confirmed their earlier conclusions and they have extended the introduction and discussion. Furthermore, they addressed all my minor issues. I have no further concerns.

One tiny suggestion is that the sentence "It is unlikely that the head-dip during inhibition of CnF vGLUT2 neurons is associated with an escape attempt since activation of CnF vGLUT2 neurons has been shown to promote escape and a robust suppression of head-dip activity in the hole-board¹²" comes before the actual results are described. I would suggest moving it to one or two sentences later.

Reviewer #1 (Remarks to the Author):

The paper by Sharma et al. investigates the functional role of a new pathway connecting the Zona Incerta (ZI) to the cuneiform nucleus (CnF) in exploration. The authors have included some new anatomical and functional studies and reorganized and improved the current version of the manuscript in response to previous comments, but there are still some problems.

The authors show that the activity of GABAergic neurons of the ZI is upregulated during rearing and during the hole-board test. They also show that photostimulation of GABAergic ZI neurons inhibits the neural activity of a small population of the CnF. They also characterize the anatomical pathway connecting the ZI to the CnF. However, there is no information about other midbrain/brainstem targets of the ZI that are mentioned in the discussion (for their role in modulating exploratory behaviors) but never in the introduction. Behavioral data supporting exploration (hole-board test and rearing) are unfortunately overshadowed by other tests (light-dark, elevated plus maze) that did not explicitly test it, and instead suggest an anxiolytic effect on behavior as mentioned earlier by the 3rd reviewer.

From the current results, we cannot exclude that the mZI-CnF pathway contributes to some of the behavioral changes in exploration, but the lack of changes in the ethograms suggest instead that this pathway plays at best a minor role.

Given the projection pattern and modulatory role of GABAergic neurons of the ZI onto the PAG and PPN, it could be important to investigate these other targets in parallel to the CnF.

Alternatively or in combination, using a retrograde photostimulation approach of CnF-projecting ZI neurons could also strengthen the authors' conclusion regarding the role of this ZI-CnF pathway in exploration.

We have discussed these concerns in the detailed responses below. To summarize the major changes to the manuscript, we have performed new experiments where we used retrograde viral targeting of the mZI - CnF circuit with photostimulation to reinforce its role in exploratory locomotion. We have included these data as part of a revised figure 7 (old figure 6). We have revised figures 1 and 6 (old figure 5) replaced the ethograms, and included more trials as suggested. We also revised figure 3, Supplementary Figure 1, and 4 (now Supplementary Figure 3). We have highlighted electrophysiology, as suggested, by moving the old Supplementary Figure 3 and incorporating it into the main text. Along with the other suggestions, this has led to a better manuscript.

Major comments

1. Glutamatergic neurons of the CnF do not only induce high-speed gait. Several studies have shown that photostimulation of glutamatergic neurons of the CnF induces a wide range of locomotor gait and speed, not only an escape or flight reaction. Chemogenetic inhibition as well as photoinhibition of glutamatergic neurons of the CnF can modulate control of slower speed. Therefore, using the rationale that the CnF only induces high-speed gaits is a misleading simplification.

We have gone through the manuscript, and this has been corrected.

2. Behavior: An increased proportion of locomotor bouts and an increased time spent in the center of the open field, as shown in Figs. 1 and 5, do not necessarily reflect an increase in exploratory behavior. Similarly, an increased activity in open arms of the elevated plus maze test or in the light chamber in the light-dark test (Fig. 2 and 6) might simply reflect a decreased anxiety in normal behaviors, thus pushing animals to expose themselves fearlessly in open areas.

These results thus do not reflect exploration, but could be considered as a prerequisite to exploration (i.e., a pre-exploratory condition). Although it has been previously reported as an escape test (as mentioned by the authors), only the hole-board test, and to some extent the rearing test, should be considered as real exploratory tasks.

The reviewer makes an astute suggestion, and we have revised our discussion to consider the possibility that decreased anxiety precedes and facilitates enhanced exploration. Our new data indicate that some anxiety tests can be mitigated independently of holeboard exploration, which may argue that they are not always linked.

The activation of inhibitory terminals in the mZI-CnF pathway (terminal stimulation and retrograde mZI-CnF stimulation) and the inhibition of CnF vGLUT2 neurons have been linked to increased exploration and decreased anxiety, as evidenced by our behavioral tests. We have emphasized the significance of these tests and incorporated additional references to bolster the discussions on anxiety and locomotion. We have reanalyzed rearing and immobility data as suggested by the reviewer (see point 3).

Direct observations from the hole-board test corroborate that inhibiting CnF vGLUT2 neurons elevates exploratory behavior. This finding is supported by the outcomes of retrograde mZI-CnF photostimulation; although the anxiety metrics were not as pronounced, increased hole-board exploration persisted.

We are grateful for the reviewer's insights, prompting us to emphasize the multifaceted roles of the CnF, including its involvement in locomotion, pain, escape behaviors, and autonomic functions.

3. Despite some statistically significant changes, the ethograms do not exhibit a clear pattern—either for the immobility in Fig. 1e-f or for the rearing in Fig. 5e-f in optogenetically manipulated mice. Overall, there is a trend with increased surveying, rearing, and locomotion upon optogenetic manipulation compared to controls in Fig. 1g or 5g. However, the effects appear to be subtle, with only a decrease in immobility upon inhibition of the CnF and an increase in rearing upon stimulation of their GABAergic inputs. These minor changes in ethograms suggest, therefore, that the ZI-CnF pathway plays a minor role in exploration.

Would it be worth analyzing several trials per mouse (rather than a single trial per mouse) and a higher number of mice to reveal stronger changes? Using naïve mice rather than accustomed

ones could also increase the probability of exploratory behaviors.

We have revised the ethograms showing more trials and have added the technical replicates to the data. The graphs in g in Figure 1 (also Figure 6) have been revised to reflect the additional data. These modifications have reinforced our original conclusion of only an effect on immobility with CNF inhibition and only an effect on rearing with mZI terminal activation. This suggests a mild effect overall on ethographic changes in the OFT. However, a common effect on exploration of the center remains for both optogenetic manipulations. In addition to the open field, our holeboard data in particular shows a robust increase in exploratory behavior.

As shown in previous studies, bilateral optogenetic manipulation might also be more efficient than unilateral stimulation in generating behavioral changes in the ethograms.

We agree that bilateral stimulation could be more effective in regards to the ethograms, but we believe that the totality of our evidence with unilateral stimulation supports our conclusion that inhibition of the CnF can promote exploratory activity. We have added a sentence about this in the discussion.

4. I appreciate the quantification of SYP-mRuby+ VGAT+ ZI boutons onto VGluT2+ CnF neurons. RNA scope is known to impair cell morphology, and I am pleased to see that the authors used DAPI to delineate the extent of the cells. Although I have a few reservations about the quantification of synaptic boutons onto largely deformed cells, I assume the analysis is representative of the observation. Nevertheless, the legend and the results section describing VGluT2 puncta and SYP-mRuby synaptic puncta quantification are confusing. I am not quite sure I understand the rationale for quantifying VGluT2 puncta in VGluT2 positive CnF neurons from the RNAscope study. What does it add to the study?

On reflection, this does not add much and have deleted h and i in figure 3. We now show the distribution of SYP-mRuby+ cells on vGLUT2+ cells in the CnF, and the proportion of positive vGLUT2+ cells, which simplifies the presentation of the data.

5. Supplementary Figure 1 raises questions about the connectome of the ZI pathway. It is still not clear whether the ZI project bilaterally or unilaterally.

Our data in supplementary Figure 1 reveal a sparse contralateral projection compared to the ipsilateral projection in the cuneiform. We have also supplied new data and revised the figure to depict the retrogradely transfected vGAT neurons bilaterally. Although some cells are visible on the contralateral side, the figure clearly shows that the main projection is ipsilateral and we have quantified this bias in a revised Supplementary Figure 1b.

According to previous tracing studies (Caggiano et al., 2018 and Dautan et al., 2021), the PPN receives far more ZI inputs compared with the CnF. And stimulation of GABAergic ZI terminals in the PPN suppresses active avoidance (Hormigo et al., 2020), thus potentially contributing to motor activity and perhaps exploration.

In Dautan et al.'s 2021 study (Figure 3), the PPN is shown to receive a notably lesser volume of inputs from ZI compared to those from CnF. This contrasts with findings by Caggiano et al. in 2018, where the PPN was observed to receive more inputs from ZI than from CnF. This discrepancy could be attributed to the number of starter cells used or inconsistencies in CnF transfection, and it doesn't necessarily indicate specific characteristics of ZI projection neurons. Recent studies have revealed a diverse range of cells within ZI, each associated with distinct functions.

Our decision not to focus on the PPN is supported by a comparison of vGLUT2 neuronal function in CnF and PPN. We elaborated in the manuscript that the activity of vGLUT2 neurons in PPN is linked to exploratory behavior, as per the findings in Caggiano's paper. Consequently, inhibiting these neurons is unlikely to affect exploratory behavior. In contrast, a decrease in exploratory activity was observed following the activation of vGLUT2 neurons in CnF. This observation positioned these cells as potential subjects for further inhibition studies.

We trust this explanation provides clearer insight into our research approach and findings.

It is also well known that the PAG is involved in strong escape and flight reactions, and it receives numerous inputs from the ZI, in addition to projections onto the CnF. Taken together, these findings suggest that the PPN and PAG could be potentially more pertinent functional targets than the CnF of GABAergic ZI neurons to exploratory behaviors.

We have emphasized in the revised discussion that the effects of mZI may involve multiple brainstem circuits. Based on the promotion of exploratory activity following vGlut2 PPN activation, we did not consider the inhibition of these cells a primary focus. Although the connectivity from mZI to PAG likely contributes, our attention was on the cuneiform as part of the MLR. Our electrophysiological data support an mZI-CnF circuit that inhibits CnF firing. The photostimulation of the retrograde mZI-CnF circuit also bolsters the evidence for our proposed circuit (see reviewer comment below).

6. I am not sure I understand why the authors cannot use a retrograde approach to stimulate ZI neurons projecting to the CnF. If photostimulation of GABAergic ZI terminals onto the CnF suggests that this pathway could be involved in exploration, there is also a high risk of activating

ZI collaterals projecting onto neighboring nuclei (i.e., the PAG and the PPN). In contrast, a retrograde approach to stimulate GABAergic ZI neurons projecting to the CnF would allow the authors to more specifically address the contribution of this pathway to exploration.

We have completed a series of experiments in which we injected a retrograde AAVrg-ChR2-DIO virus into the cuneiform. These experiments yielded interesting findings that complement and reinforce our previous data, and we thank the reviewer for this suggestion. These new data are illustrated in a revised Figure 7 in the main text, and the anatomical details are captured in a revised Supplementary Figure 1. Using this approach, we labelled a subpopulation of mZI neurons and compared the effects to those observed with the anterograde approach. The impact on exploratory locomotion persisted, but the anxiolytic effects were less robust.

We suspect a threshold effect may be in play since the labeled GABAergic neuron population was smaller with the retrograde approach, and therefore no effect was observed in the EPM. The new data presented in this revision at the reviewer's suggestion further supports the conclusion that exploratory locomotion is the dominant effect of the mZI-CnF pathway. Notably:

1. Our fiber photometry results captured a population of GABAergic neurons involved in exploratory behaviour, with dominant peaks observed during holeboard investigation. Conversely, our ephys data showed a spike reduction in CnF when activating a pool of GABA neurons in mZI.
2. Combining the results from anterograde terminal stimulation of GABA ZI terminals in CnF, photoinhibition of CnF vGLUT2 neurons, and retrograde CnF to mZI stimulation, we observed consistent support for an increase in exploratory locomotion.

Minor comments

Introduction

Page 1, line 7, replace with more recent reviews.

Done

Page 1, 2nd paragraph

Replace the reviews with scientific articles in "The CnF has connectivity with the sympathetic system."

Done

Page 1, 1st paragraph

CnF neurons are also involved in walking gaits, not just high-speed locomotion (Fig. 1J of Caggiano et al., 2018, and Fig .6C of Josset et al., 2018). CnF is not incompatible with slower locomotion, and this should be clearly stated. Same for the discussion at page 7.

We agree with this and have explicitly stated so in the revised manuscript; however, the results showing CnF vGLUT2 inhibition suggest sufficiency, not necessity. The fact that high-speed

locomotion is blocked while lower speed gaits (27 cm/s to 20 cm/s) are spared highlights that these neurons are not necessary for slower-speed locomotion but are required for high-speed locomotion.

Figure 1

Ethograms in Figure 1e-f. The main finding regarding behavior in Fig 1g is less immobility. When looking at the ethograms, one mouse is mostly immobile. One clear thing is the lack of effect on behavior, which is confirmed in Fig. 1g. This is odd because mice show more locomotor bouts when CnF is inhibited (Fig. 1i). Also, what is shown in those ethograms, a single trial for each mouse? What is the bin size? Please state these details in the legend. Have you performed several trials? If yes, what is the proportion of each behavior?

We have reorganized Figure 1. The ethograms have now been expanded as suggested with the total number of trials specified in the legend. With the reanalysis, the discrepancies in behavior observed in panels i and g have been resolved. Previously, panel g reported a single trial per mouse across multiple mice; it now presents group means of three replicates per mouse. We applied similar modifications to Figure 6 (formerly Figure 5) for consistency. In terms of behavior the effects on OFT are mild, especially on locomotor activity. Interestingly, we did not find any association between locomotion velocity and Ca²⁺ transients during the OFT (Figure 5l-n).

Page 2, line 2

This study is a non-peer-reviewed article and it does not show any evidence of interconnection between both nuclei. I suggest replacing the reference with a peer-reviewed paper and/or correct the statement about connectivity throughout the text.

Deleted this reference

Page 3, 1st paragraph

Several statements without statistical analysis should be removed from the paper.

“During spontaneous exploration... largely restricted to walking and trotting gaits.”

“We observed that control... showed a time-dependent reduction in locomotion.”

In absence of gait and statistical analyses, these are only qualitative and speculative.

Deleted

Page 3, 3rd paragraph

“with the exception of few studies... head-dip activity in the hole-board.”

The interpretation of the hole-board test as an exploratory test and its potential assessment as a measure of escape should be discussed in the discussion rather than in the results section.

Moved to discussion.

Page 4, conclusion at the top. Unilateral inhibition of the CnF does not suppress locomotor behavior according to your data. Remove or rephrase.

We have rephrased this sentence.

Supplementary Figure 1

Without quantification, it is difficult to concur with a unilateral bias. Furthermore, it shows a labelling as dense in the PAG and PPN, supporting the need to investigate these pathways. We have updated the image to zoom in on the CnF, PAG, and PPN. Our initial focus was on projections onto the CnF, using RNAscope. For technical reasons we could not reimage the slides to collect the PPN and PAG contralateral data. We agree with the reviewer that examining bilateral projections is an important aspect, and we did see qualitatively more fibres projecting toward the ipsilateral CnF in Supplementary Figure 1a. We resolved this point with additional experiments in which we injected the retrograde AAVrg virus unilaterally into the CnF and showed sparse contralateral labelled somata compared to the ipsilateral side (Supplementary figure 1b - new data). We quantified these retrograde data. We have updated Supplementary Figure 1 to include our new anatomical data and accompanying quantification.

Page 4. "Our data also shows that mZI GABAergic projections are biased towards a subpopulation of CnF vGLUT2 neurons (Figure 3i)." Maybe rephrase because I do not see this bias. Also, it seems arbitrary to use these three categories of puncta density. Why not simply report the distribution of puncta per cell?

Per the reviewer's suggestion, we have simplified the figure to remove e,h, and i and only include graphs f and g, now figure 1e and f, respectively. The text relating to these panels has been removed since the panel was deleted. We have also rephrased the results section to clarify that a majority of cells positive for SYP-mRUBY GABAergic puncta are also vGLUT2+ and that within the total CnF vGLUT2+ population, there is a subpopulation of cells associated with SYP-mRUBY+ synaptic puncta. Puncta categories are now condensed to only indicate positive and negative expression. The thresholds for both SYP and vGLUT2 (≥ 4 puncta) are detailed on page 15 of the methods section.

Page 4, line 1,

Again, correct the statement: the CnF is not only correlated with high-speed locomotion.

Corrected

Page 4, 1st paragraph

"Monosynaptic retrograde... neurons¹⁴." Add Caggiano et al., 2018 as a reference.

Corrected

Page 4, 2nd paragraph

The verb "to demonstrate" is used several times throughout the draft. It is improper in the current context. Please correct.

We have changed the wording.

"A recent study...vglut2 mRNA puncta" should be moved to the discussion.

Moved to discussion

Figure 3

The heatmap of reporter mGFP is not necessary in Fig. 3b.

There is a high density of ZI terminals onto the PAG and a smaller patch in the CnF in Fig. 3c. It would be great to show the axonal terminal pattern of ZI onto the PAG, CnF, and PPN in more rostral sections. Quantification will be important.

These are all great points. The heatmap in Fig. 3b has been removed. We acknowledge the expression pattern from the mZI to the PAG, and discuss the potential for mZI projections to contribute to changes in behavior. Our focus was on the cuneiform, as the mZI to the cuneiform circuit has been largely unexplored. We wished to investigate how manipulating mZI-cuneiform circuits could alter exploratory behavior. Unfortunately, the rostral sections of the cuneiform in the original set of experiments were not focused on in the original set of experiments and we do not have the tissue.

Figure 4.

Open field locomotion and ZI activation are not related but the current z-score scaling overemphasize the small 0.2 peak at the onset of locomotion. I suggest that the authors match the scaling of panel n with the one of panel j for better clarity of results.

Agree. This has been done.

Page 5, 1st paragraph

Supplementary Figure 3 on multi-unit recordings of CnF neurons upon photostimulation of the VGAT+ ZI neurons adds to the comprehension of the results, but it should be integrated into the paper and better presented. It is not clear from the results section how many cells were recorded from 3 mice. The firing frequency of CnF neurons, the total number of recorded multi-units, and the number of depressed cells upon ZI photostimulation should be reported. In panel 3c, why is there a category for "increase" in the legend but not in the pie chart itself? What are the statistical criteria mentioned in the methodology to define a responding (inhibited) cell? One can clearly see inhibition, but there is no clear evidence of a post-inhibitory rebound activity.

We have included experimental details in the results section for this experiment. The label for 'increase' is included for completeness, even though no cells exhibited increased firing rates; we have clarified this in the figure caption. We have also elaborated on the statistical criteria for inclusion in the methods section. The discussion regarding post-inhibitory rebound has been omitted, and the figure, previously part of the supplementary data, is now incorporated into the main paper (Figure 4).

Page 5, 2nd paragraph

Add Caggiano et al., 2018 as ref to ref 46-47 about head-dipping.

Added.

The increased calcium activity in VGAT ZI neurons during rearing and head-dipping is surprising and weird. How can the same neuronal subpopulation generate at once a similar activity for two

distinct exploratory behaviors? It is not clear to me.

Thank you for this - it is a great point. Because we are recording the entire labeled population, subpopulations could produce distinct behaviors. To resolve this would require a miniscope experiment, which is beyond the scope of this paper. We have added a sentence in the discussion to address this point.

Some interpretations of the results are speculative and should be transferred to the discussion Page 5, "Furthermore.... ...during escape behavior.21"

Removed

Page 6, 1st paragraph, "These results highlight a role of ZI GABA... ... novelty-seeking.52"

Removed

Figure 5

In comparison to supp. Figure 1, I am a bit surprised by the low expression of terminals in the CnF and the absence of ChR2-YFP expression in the PAG or PPN from this tracing in Figures 1 and 5. The tip of the optical fiber is far away from the ChR2-YFP expressing ZI terminals in the CnF in Fig. 5. Was it sufficient to activate GABAergic terminals?

Fig. 5e does not show any striking pattern, despite statistical differences in rearing in Fig. 5g.

Figure 1b shows the somatic expression of the virus within the CnF. This is a vGlut2-Cre mouse targeted with AAV-DIO-eNpHR3.0-eYFP. We have reimaged the figures in 1b and 5b (now figure 6b) to be consistent with supp figure 1. The tip of the optrode is closer than appreciated from the old figure. We replaced them with extended ethograms, and we have added the number of trials per mouse. Overall, the pattern in the ethograms matches our data from 5g, concluding mild changes in the open field.

Although CnF-projecting VGAT+ZI terminals do not affect immobility, they seem to increase motor activity (i.e., rearing). It seems that they also tend to increase locomotor activity. Maybe when assessing more than one trial per mouse these effects could be significant?

We have included multiple trials, and the rearing is significant. The other measures are not. This is perhaps not too surprising as it mirrors the effects found with inhibition of the CnF vGLUT2 neurons. Overall, it suggests that the effects in the open field are generally mild - with an increase in rearing being the metric most affected. Movement into the centre suggests both an anxiolytic and exploratory locomotion effect.

The increased trend in travelled distance and time upon stimulation of VGAT+ZI terminals of YFP expressing mice in Fig. 5j-k is a bit puzzling. This raises questions about the efficacy of blue light to induce behavioral changes in wild-type mice in the absence of ChR2. Fig. 6 is more convincing than Fig. 5a-g by showing a clear effect in the elevated plus maze test, the light-dark test, and the hole-board test with a good control group showing very few changes upon stimulation.

We speculate that the increase in center exploration observed in the eYFP control group is a consequence of greater habituation to the environment in this cohort rather than a consequence of blue light stimulation, as the OFF period saw elevated center exploration in this group. The same cohort of mice in Figure 5 (now Figure 6) was analyzed on the EPM/light-dark/hole-board tests in Figure 6 (now Figure 7), which suggests this is not a consequence of light-evoked behavior in the control group.

Supplementary Figure 1

The outline should be aligned with the brain section image. There is no clear difference at first sight in the axonal terminal patterns onto the CnF, PPN, and PAG. There is still a question about the lateralization of this ZI pathway. Some quantification will be necessary.

The figure outline has been updated and aligned. In our examination of the retrograde effectiveness of mZI stimulation, we injected the retrograde AAV virus unilaterally into the CnF. We then presented the ipsi- and contralateral somas from the mZI in a revised figure. The difference is striking, and we have provided quantification as requested.

Supp. Fig. 4 is not necessary and misleading as the CnF does not generate only escape or flight behavior.

We have deleted the figure

Discussion

Page 7

Add Caggiano et al., 2018 to ref 14.

Added

Correct last sentence in regard to the CnF or PPN modulation of slow locomotion (Caggiano et al., 2018).

Need to check

Page 8, 1 line, correct the statement

Corrected

Page 8, 3rd paragraph should be reorganized; studies that investigated PPN and PAG populations receiving ZI inputs should be better presented and in context with the current study (Ferreira-Pinto et al., 2021; Chou et al., 2018). The use of the MLR instead of the PPN regarding the Ferreira-Pinto et al., 2021 study is misleading and could give the impression that they studied distinct CnF-related pathways, while it was actually the PPN.

We have reorganized this paragraph.

Page 8: "These observations suggest that CnF inhibition contributes to postural adjustments and the suppression of competing motor pathways (e.g.: walking, rearing, grooming, etc.)."

There are no statistical changes to support this conclusion.

We deleted this sentence.

Page 9: "...mZI terminals could activate neighboring nuclei..." Do you mean inhibit?
Thank you - now corrected

Reviewer #2 (Remarks to the Author):

The authors addressed all issues I raised about adding several experiments. However, I think that the authors need to edit some parts of the main text for clarifying some points as below.

1. In the last paragraph of page 2, I cannot understand why the authors put the sentence 'The behavioral repertoire of mice is sensitive to environmental features' in the middle of this paragraph.

Agreed - this is more of a discussion item. We have deleted it.

2. In the second paragraph of page 3, the authors need to add some words after 'To confirm' to clarify the purpose of this experiment

We revised the sentence to clarify.

3. In page 3, the third paragraph is not clear

Agreed. We have rewritten it to make it clear - see also Reviewer 3 comments.

4. In the first paragraph of page 6, I think that the authors cannot say that mZI GABAergic neurons modulate fear generalization, anxiety, and novelty-seeking in this paragraph. They just show the neural correlates of mZI inhibitory vGAT neurons

Agreed - we have deleted this - see also Reviewer 1.

Reviewer #3 (Remarks to the Author):

The authors have done extra experiments, which all confirmed their earlier conclusions and they have extended the introduction and discussion. Furthermore, they addressed all my minor issues. I have no further concerns.

One tiny suggestion is that the sentence "It is unlikely that the head-dip during inhibition of CnF vGLUT2 neurons is associated with an escape attempt since activation of CnF vGLUT2 neurons has been shown to promote escape and a robust suppression of head-dip activity in the hole-board¹²" comes before the actual results are described. I would suggest moving it to one or two sentences later.

Thank you and we have moved the sentence.

REVIEWERS' COMMENTS

Reviewer #1 (Remarks to the Author):

The draft is clearly improved, nevertheless, some issues still need to be fixed. The use of acronyms for some tests (OFT, EPM) is inconsistent through the text. I think it would look better to state the test's name plainly each time. The description of the tracing experiments could be improved. Ethograms are still not clear: there are no clear signals from the examples from Fig. 1 and 6. There is a significant increase in in Fig. 6g, but we do not see anything in Fig. 6e-f. The signal is very noisy. Presenting all animals in a single matrix is not helpful, data are diluted, and no conclusions seem to emerge. Maybe presenting a typical example in the main figure and all individual ethograms in supplementary figures might fix the problem.

Minor comments

Introduction

Page 2

"The medial zona incerta (mZI) is located on the border of the rostral and caudal ZI, encompassing part of the rostral and caudal quadrants."

This is a bit repetitive.

Page 3

"Photoinhibition of CnF glutamatergic vGLUT2 neurons significantly increased immobility bouts in vGLUT2eNpHR3.0 mice during laser ON epoch (Figure 1g)."

Figure 1g reports a decrease in immobility duration, not in immobility bouts. Moreover, the significance seems to rely on a single outlier. What about the stats if the outlier is removed? Please correct.

Legend Fig. 1g: Panels do not seem to show total duration, but percentage change of total duration. Please clarify.

"activation of the CnF vGLUT2 neurons can produce a greater range of locomotor speed"

Add "s" to speed. To be consistent through the paragraph, add ref 14,15,16.

"PPN vGLUT2 neurons are important for slower gait locomotion, independent of CnF vGLUT2 neurons¹⁷. Highlighting PPN heterogeneity, the effects of PPN vGLUT2 photoactivation have been associated with slowing or stopping of locomotion^{15,16}. In contrast, activation of the CnF vGLUT2 neurons can produce a greater range of locomotor speed."

Without experiments targeting the PPN, this statement should be in the discussion.

Figure 4a: how did the authors identify anatomically recorded neurons?

Page 4

mZI acts as a source of inhibitory projections onto CnF vGLUT2 neurons

This section needs a bit of editing. Some information is not relevant, and some statements raise questions that should be addressed in the discussion, not in the results.

"The mZI is a region of interest because the ZI is recognized as a hub for sensorimotor integration and is associated with the genesis of complex locomotor behaviors^{34-36,39,45}."

This is in the middle of the paragraph after presenting the retrograde tracing experiment. Remove it.

"Confirming a mainly ipsilateral projection, retrograde tracing of the mZI-CnF circuit revealed sparse cell bodies contralaterally (Supplementary Figure 1b)."

Why state this at the end, while the data from the retrograde tracing are presented at the beginning of the paragraph.

" SYP-mRUBY+ synaptic puncta were evaluated across all vGLUT2+ cells within the CnF. Our results suggest that there is a subpopulation of CnF vGLUT2+ cells that are associated with SYP-mRUBY+ synaptic puncta (Figure 3f)."

The results do not suggest, they show it..

“the mZI led to a inhibition”

An inhibition.

Page 5

Figure 5d-f: What is the threshold for calcium transients to be considered significant? The threshold is 2.91 median absolute deviation from the median according to the methodology, but examples are far from these values. Although the Z-scores in Fig. 5i-j are clearer, they are still below the threshold.

This need to be clarified, and a reference line for the threshold should be added to the figures.

“inhibition of firing frequency at monosynaptic latencies in a subpopulation of CnF cells”

How did the authors asses monosynaptic latency? The latencies are not shown in figures or reported in the text. Maybe refer to short latency.

Page 6

mZI promotes exploratory behavior via inhibitory projections to the CnF

“However, we observed a significant increase in both the center distance traveled (Figure 6j) and the time spent in the center zone of the open field in vGATChr2 mice during photostimulation (Figure 6k).”

With such an overlap in standard deviations, it is hard to believe that it is significantly different. Presenting standard deviation while running a non-parametric test is confusing at best. Maybe consider using boxplots or violin plots.

In addition, the authors state in the discussion on page 9: “the more robust effects of the mZI vGAT photostimulation compared to CnF vGLUT2 inhibition.” The degree of significance supports this; however, the current presentation using mean and SD does not.

Data from Fig. 7 should be presented in a different paragraph.

Page 7

“To further support that mZI-CnF circuit contributed to exploratory locomotor behavior, we injected a retrograde AAVrg-DIO-ChR2-eYFP into vGAT-IRES-Cre mice and ”

Please specify that the AAV injection was in the CnF.

“Upon photostimulation of the Chr2 in retrogradely labeled mZI neurons, we found that the EPM data showed no significant change in behavior (Figure 7f-g). In the light-dark transition test, mice spent more time in the light chamber when photostimulated (Figure 7k), while the time per visit remained unchanged (Figure 7l). Finally, the head-dips (Figure 7p) and duration of head-dips (Figure 7q) were significantly increased in the hole-board test.”

The data are not significant in Fig. 7k during photostimulation, only after. Please correct.

“Overall, photostimulation of the mZI GABAergic neurons projecting to the CnF produced similar but attenuated results as seen in the mZI-CnF GABAergic terminal photostimulation experiments.” Alternatively, results in the Elevated Plus Maze and Light-dark test from anterograde AAV suggest that activation of the terminals recruited a greater number of targets (presumably the PAG and the PPN), whereas the only effects in the hole board test upon activation of the retrograde AAV might suggest a more specific recruitment of mZI neurons involved in exploration.

Discussion

Page 7

“inhibition of firing rates”

Rephrase to decreased firing rate or inhibition of firing.

Page 8

“hunger is known to promote exploratory behavior and foraging”

Add a reference.

Page 9

“One distinct subpopulation with descending connectivity to the spinal cord can control vertical movement during rearing¹⁸”

For clarity, specify that these are PPN neurons.

“While it is possible that this inhibition is indirect, the latency of the depression is consistent with monosynaptic connectivity³⁶.”

This reference refers to the connectivity between ZI and PAG neurons, not the CnF. Maybe simply remove the reference.

“This could partly explain the more robust effects of mZI photostimulation compared to CnF vGLUT2 inhibition”

I am not sure I understand: results from inhibition of CnF VGLUT2 neurons in Fig. 2 and activation of mZI vGAT neurons in Fig. 7 are very similar. This goes back to the point raised about the Results section.

“Our findings and that of others raise an interesting possibility that the CnF, which is predominantly associated with motor function^{16,18,59}”

Add Caggiano et al., 2018 and Josset et al., 2018 as references.

The discussion is still very long.

Ref 59 is no longer a pre-print, it is published.

REVIEWERS' COMMENTS

Reviewer #1 (Remarks to the Author):

The draft is clearly improved, nevertheless, some issues still need to be fixed. The use of acronyms for some tests (OFT, EPM) is inconsistent through the text. I think it would look better to state the test's name plainly each time. The description of the tracing experiments could be improved.

We have fixed acronyms used to make them consistent. The tracing experiment paragraph has been revised.

Ethograms are still not clear: there are no clear signals from the examples from Fig. 1 and 6. There is a significant increase in Fig. 6g, but we do not see anything in Fig. 6e-f. The signal is very noisy. Presenting all animals in a single matrix is not helpful, data are diluted, and no conclusions seem to emerge. Maybe presenting a typical example in the main figure and all individual ethograms in supplementary figures might fix the problem.

Thank you for this comment. We have taken the reviewer's suggestion to focus on particular behaviours - namely, those that are significant. For Figure 1, we have updated the ethogram to include immobility data only. For Figure 6, we have included immobility and rearing. This has made the ethograms easier to read, and the critical data are now illustrated. We have added the non-significant ethograms as separate supplementary figures (Supplementary Figures 1 and 4).

Minor comments

Introduction

Page 2

"The medial zona incerta (mZI) is located on the border of the rostral and caudal ZI, encompassing part of the rostral and caudal quadrants."

This is a bit repetitive.

Removed.

Page 3

"Photoinhibition of CnF glutamatergic vGLUT2 neurons significantly increased immobility bouts in vGLUT2eNpHR3.0 mice during laser ON epoch (Figure 1g)."

Figure 1g reports a decrease in immobility duration, not in immobility bouts. Moreover, the significance seems to rely on a single outlier. What about the stats if the outlier is removed? Please correct.

Corrected in the text on page 3. The figure 1g legend was correct.

The stats were still significant with the outlier removed ($p = .0115$). We did make changes to the averaging of the data, with it now referred to pre-stim for each trial. This was necessary since the original data referred to the baseline condition, not the adjacent pre-stim data, and was not reflect the changes observed in the replotted ethograms (see comment on ethograms above). Similar changes were made to Figure 6 to remain consistent.

Legend Fig. 1g: Panels do not seem to show total duration, but percentage change of total duration. Please clarify.

We have clarified in the legend that the total duration of behaviors during each stimulation trial was corrected to baseline pre-stim conditions.

“activation of the CnF vGLUT2 neurons can produce a greater range of locomotor speed”

Add “s” to speed. To be consistent through the paragraph, add ref 14,15,16.

This sentence has been deleted as per the comment below.

“PPN vGLUT2 neurons are important for slower gait locomotion, independent of CnF vGLUT2 neurons¹⁷. Highlighting PPN heterogeneity, the effects of PPN vGLUT2 photoactivation have been associated with slowing or stopping of locomotion^{15,16}. In contrast, activation of the CnF vGLUT2 neurons can produce a greater range of locomotor speed.”

Without experiments targeting the PPN, this statement should be in the discussion.

This has been repeated in the discussion, so to be concise, we have removed it from the results section as suggested.

Figure 4a: how did the authors identify anatomically recorded neurons?

The stereotaxic location of each recording site relative to the surface of the brain was recorded during the experiment. To verify this estimate post hoc, the deepest location of each recording tract was lesioned, and the location of each recording was revised relative to this histologically verified final location of the probe. This text has been added to page 12 of the manuscript.

Page 4

mZI acts as a source of inhibitory projections onto CnF vGLUT2 neurons

This section needs a bit of editing. Some information is not relevant, and some statements raise questions that should be addressed in the discussion, not in the results.

“The mZI is a region of interest because the ZI is recognized as a hub for sensorimotor integration and is associated with the genesis of complex locomotor behaviors^{34–36,39,45}.”

This is in the middle of the paragraph after presenting the retrograde tracing experiment. Remove it.

We have removed it from the results section as suggested and added it to the introduction.

“Confirming a mainly ipsilateral projection, retrograde tracing of the mZI-CnF circuit revealed sparse cell bodies contralaterally (Supplementary Figure 1b).”

Why state this at the end, while the data from the retrograde tracing are presented at the beginning of the paragraph.

Corrected as suggested.

“ SYP-mRUBY+ synaptic puncta were evaluated across all vGLUT2+ cells within the CnF. Our results suggest that there is a subpopulation of CnF vGLUT2+ cells that are associated with SYP-mRUBY+ synaptic puncta (Figure 3f).”

The results do not suggest, they show it...

Corrected in the text. Also, we simplified the graph in Figure 3e, amalgamating 3e and f into one graph.

“the mZI led to a inhibition”

An inhibition.

Corrected in the text.

Page 5

Figure 5d-f: What is the threshold for calcium transients to be considered significant?

The threshold is 2.91 median absolute deviation from the median according to the methodology, but examples are far from these values. Although the Z-scores in Fig. 5i-j are clearer, they are still below the threshold.

This need to be clarified, and a reference line for the threshold should be added to the figures.

Thank you for these comments. We did not select traces based on threshold. Traces were averaged and triggered from the start of behaviors. We have removed the sentence “Peak thresholds were calculated by setting threshold values as 2.91 median absolute deviations from the median” as this is misleading. We have quantified the data to Figure 5d, and j to address the reviewer’s comments.

“inhibition of firing frequency at monosynaptic latencies in a subpopulation of CnF cells”
How did the authors assess monosynaptic latency? The latencies are not shown in figures or reported in the text. Maybe refer to short latency.

We have reworded in two sections (page 4, results; page 9, discussion) to refer to short rather than monosynaptic latencies. We have described this short latency as <100ms, which was the bin size used in the electrophysiological analysis. Any more detailed analysis of response latency would require IPSC recordings.

Page 6

mZI promotes exploratory behavior via inhibitory projections to the CnF

“However, we observed a significant increase in both the center distance traveled (Figure 6j) and the time spent in the center zone of the open field in vGATChR2 mice during photostimulation (Figure 6k).”

With such an overlap in standard deviations, it is hard to believe that it is significantly different. Presenting standard deviation while running a non-parametric test is confusing at best. Maybe consider using boxplots or violin plots.

In addition, the authors state in the discussion on page 9: “the more robust effects of the mZI vGAT photostimulation compared to CnF vGLUT2 inhibition.” The degree of significance supports this; however, the current presentation using mean and SD does not.

The data were significant since they refer to the increase from baseline. The increase was significant for ChR2 but not for eYFP. We have clarified this in the text.

Data from Fig. 7 should be presented in a different paragraph.

Done

Page 7

“To further support that mZI-CnF circuit contributed to exploratory locomotor behavior, we injected a retrograde AAVrg-DIO-ChR2-eYFP into vGAT-IRES-Cre mice and “
Please specify that the AAV injection was in the CnF.

Corrected in the text.

“Upon photostimulation of the ChR2 in retrogradely labeled mZI neurons, we found that the EPM data showed no significant change in behavior (Figure 7f-g). In the light-dark transition test, mice spent more time in the light chamber when photostimulated (Figure 7k), while the time per visit remained unchanged (Figure 7l). Finally, the head-dips (Figure 7p) and duration of head-dips (Figure 7q) were significantly increased in the hole-board test.”

The data are not significant in Fig. 7k during photostimulation, only after. Please correct.

Corrected in the text.

“Overall, photostimulation of the mZI GABAergic neurons projecting to the CnF produced similar but attenuated results as seen in the mZI-CnF GABAergic terminal photostimulation experiments.”

Alternatively, results in the Elevated Plus Maze and Light-dark test from anterograde AAV suggest that activation of the terminals recruited a greater number of targets (presumably the PAG and the PPN), whereas the only effects in the hole board test upon activation of the retrograde AAV might suggest a more specific recruitment of mZI neurons involved in exploration.

We have adapted the following text in the discussion. “Although the optic probe was centered over the CnF, where functional connections were found, activation of mZI terminals could inhibit neighboring nuclei, such as the PAG or PPN. We did tackle this issue with retrograde targeting of the mZI-CnF circuit which produced more targeted effects on holeboard exploration. This may suggest a more specific circuit from the mZI to the CnF targeting exploratory behavior”

Discussion

Page 7

“inhibition of firing rates”

Rephrase to decreased firing rate or inhibition of firing.

Rephrased to inhibition of firing as suggested.

Page 8

“hunger is known to promote exploratory behavior and foraging”

Add a reference.

Added the following paper reference in the text.
“Hunger-driven adaptive prioritization of behavior
Smith NK, Grueter BA”

Page 9

“One distinct subpopulation with descending connectivity to the spinal cord can control vertical movement during rearing¹⁸”

For clarity, specify that these are PPN neurons.

Done

“While it is possible that this inhibition is indirect, the latency of the depression is consistent with monosynaptic connectivity³⁶.”

This reference refers to the connectivity between ZI and PAG neurons, not the CnF. Maybe simply remove the reference.

Done

“This could partly explain the more robust effects of mZI photostimulation compared to CnF vGLUT2 inhibition”

I am not sure I understand: results from inhibition of CnF VGLUT2 neurons in Fig. 2 and activation of mZI vGAT neurons in Fig. 7 are very similar. This goes back to the point raised about the Results section.

We have removed this section, as it was repetitive.

“Our findings and that of others raise an interesting possibility that the CnF, which is predominantly associated with motor function^{16,18,59}”

Add Caggiano et al., 2018 and Josset et al., 2018 as references.

This has been completed.

The discussion is still very long.

We have removed certain repetitive paragraphs along with the discussion of possible effects on the cardiovascular system. Overall we have reduced the word count from 1801 words to 1566 words.

Ref 59 is no longer a pre-print, it is published.

Done